# Structure of a monomeric photosystem I core associated with iron-stress-induced-A proteins from *Anabaena* sp. PCC 7120

Ryo Nagao [1,8,12] ✉, Koji Kato[1,9,12], Tasuku Hamaguchi[2,10,12], Yoshifumi Ueno[3,11], Naoki Tsuboshita[1], Shota Shimizu[1], Miyu Furutani[3], Shigeki Ehira[4], Yoshiki Nakajima [1], Keisuke Kawakami [2], Takehiro Suzuki[5], Naoshi Dohmae [5], Seiji Akimoto [3] ✉, Koji Yonekura [2,6,7] ✉ & Jian-Ren Shen [1] ✉

Iron-stress-induced-A proteins (IsiAs) are expressed in cyanobacteria under iron-deficient conditions. The cyanobacterium *Anabaena* sp. PCC 7120 has four *isiA* genes; however, their binding property and functional roles in PSI are still missing. We analyzed a cryo-electron microscopy structure of a PSI-IsiA supercomplex isolated from *Anabaena* grown under an iron-deficient condition. The PSI-IsiA structure contains six IsiA subunits associated with the PsaA side of a PSI core monomer. Three of the six IsiA subunits were identified as IsiA1 and IsiA2. The PSI-IsiA structure lacks a PsaL subunit; instead, a C-terminal domain of IsiA2 occupies the position of PsaL, which inhibits the oligomerization of PSI, leading to the formation of a PSI monomer. Furthermore, excitation-energy transfer from IsiAs to PSI appeared with a time constant of 55 ps. These findings provide insights into both the molecular assembly of the *Anabaena* IsiA family and the functional roles of IsiAs.

Oxygenic photosynthesis of cyanobacteria, various algae, and land plants converts light energy from the sun into biologically useful chemical energy concomitant with the evolution of molecular oxygen[1]. The central part of the light-energy conversion is two multi-subunit pigment-protein complexes, photosystem I and photosystem II (PSI and PSII, respectively), which perform light-driven charge separation and a series of electron-transfer reactions[1]. Among these complexes, PSII organizes mainly into a dimer regardless of species of the organism[2,3], whereas PSI exhibits different structural organization among photosynthetic organisms[4–6]. Prokaryotic cyanobacteria have

trimeric[7–10] or tetrameric PSIs[10–14] in addition to other minor forms of PSI monomers and dimers[15,16].

Iron is essential for photosynthetic organisms and involved in electron-transfer reactions, although it is a scarce component in the ocean and fresh-water environment[17]. When cyanobacteria are exposed to iron deficiency, they express iron-stress-induced-A proteins (IsiAs)[18–20] encoded by the *isiA* genes, which are one of the membrane-embedded light-harvesting complexes (LHCs). Structural studies have revealed that up to 18 copies of IsiA encoded by one *isiA* gene surround a trimeric PSI core, forming a PSI-IsiA supercomplex

[1]Research Institute for Interdisciplinary Science and Graduate School of Natural Science and Technology, Okayama University, Okayama 700-8530, Japan. [2]Biostructural Mechanism Laboratory, RIKEN SPring-8 Center, Hyogo 679-5148, Japan. [3]Graduate School of Science, Kobe University, Hyogo 657-8501, Japan. [4]Department of Biological Sciences, Graduate School of Science, Tokyo Metropolitan University, Tokyo 192-0397, Japan. [5]Biomolecular Characterization Unit, RIKEN Center for Sustainable Resource Science, Saitama 351-0198, Japan. [6]Institute of Multidisciplinary Research for Advanced Materials, Tohoku University, Miyagi 980-8577, Japan. [7]Advanced Electron Microscope Development Unit, RIKEN-JEOL Collaboration Center, RIKEN Baton Zone Program, Hyogo 679-5148, Japan. [8]Present address: Faculty of Agriculture, Shizuoka University, Shizuoka 422-8529, Japan. [9]Present address: Structural Biology Division, Japan Synchrotron Radiation Research Institute (JASRI), Hyogo 679-5198, Japan. [10]Present address: Institute of Multidisciplinary Research for Advanced Materials, Tohoku University, Miyagi 980-8577, Japan. [11]Present address: Institute of Arts and Science, Tokyo University of Science, Tokyo 162-8601, Japan. [12]These authors contributed equally: Ryo Nagao, Koji Kato, Tasuku Hamaguchi. ✉e-mail: nagao.ryo@shizuoka.ac.jp; akimoto@hawk.kobe-u.ac.jp; yone@spring8.or.jp; shen@cc.okayama-u.ac.jp

with a closed ring of IsiAs in many cyanobacteria[21–25], where IsiA can function to donate excitation energy to PSI[25–28]. IsiAs have been observed to associate with PSI but not PSII, hence contribute to the photochemical reactions of PSI under iron-deficient conditions in various cyanobacteria.

A very attractive feature of the IsiA family is that the number of *isiA* genes differs among cyanobacteria. The cyanobacterium *Leptolyngbya* sp. strain JSC-1 (hereafter referred to as *Leptolyngbya*) has five *isiA* genes of *isiA1–5*, all of which were expressed under an iron-deficient condition[29]. Our phylogenetic analysis proposed that the cyanobacterium *Anabaena* sp. PCC 7120 (hereafter referred to as *Anabaena*) has four types of the *isiA* genes: *isiA1*, *isiA2*, *isiA3*, and *isiA5*, based on the similarities of the sequences to *Leptolyngbya*[30]. Our previous study also showed that the IsiA protein forming the ring structure surrounding the PSI-core trimer in *Synechocystis* sp. PCC 6803, *Synechococcus elongatus* PCC 7942, and *Thermosynechococcus vulcanus* NIES-2134[23–25] is similar to the *isiA1* gene of *Anabaena*[30]. Two-dimensional blue-native (BN)/SDS-PAGE analysis using thylakoids from the cells grown under the iron-deficient condition detected IsiA in a PSI fraction, which was located near the band of the PSII dimer, showing the formation of a PSI-IsiA supercomplex in *Anabaena*[30]. Since *Anabaena* has not only tetrameric PSI cores[11–13] but also PSI monomers and dimers[15], it was suggested that the supercomplex was composed of either a PSI core monomer with several IsiA subunits or a PSI dimer with a few IsiAs based on the putative molecular weight of PSI-IsiA[30]. These observations raise two issues as to (1) how the PSI-IsiA supercomplex is organized, and (2) why the IsiA subunits are not associated with the PSI tetramer.

In this study, we solved a structure of a PSI-IsiA supercomplex purified from *Anabaena* grown under the iron-deficient condition by cryo-electron microscopy (cryo-EM) single-particle analysis. The results showed the existence of a PSI-IsiA supercomplex consisting of six IsiA subunits encoded by different *isiA* genes associated with a monomeric PSI core, forming a PSI-monomer-IsiA supercomplex. We reveal and discuss the expression of the different *isiA* genes, the association pattern of each IsiA with the PSI core, and their roles in energy transfer.

## Results

### Expression and accumulation of IsiAs
The expression of the *isiA* genes in *Anabaena* was examined by qRT-PCR (Supplementary Fig. 1a), which showed that the transcript levels of three *isiA* genes, *isiA1*, *isiA3*, and *isiA5*, are markedly increased under the iron-deficient condition, whereas the transcript level of *isiA2* is increased to a less remarkable level. The PSI-IsiA supercomplexes were purified by trehalose density gradient centrifugation (Supplementary Fig. 1b; see "Methods"), which showed that the supercomplex contains all of the four IsiAs (Supplementary Fig. 1c). Absorption and fluorescence spectra, and pigment compositions of the purified supercomplex are summarized in Supplementary Fig. 1d–f, which showed that the supercomplex is characteristic of a PSI-like preparation in terms of the absorption and fluorescence bands, and contains chlorophylls (Chls) *a*, *β*-carotenes and a small amount of echinenones.

### Overall structure of the PSI-IsiA supercomplex
Cryo-EM images of the PSI-IsiA supercomplex were obtained by a JEOL CRYO ARM 300 electron microscope operated at 300 kV. After processing of the images with RELION (Supplementary Fig. 2, Supplementary Table 1), the final cryo-EM map was determined with a C1 symmetry at a resolution of 2.62 Å, based on the "gold-standard" Fourier shell correlation (FSC) = 0.143 criterion (Supplementary Fig. 3a), although the peripheral region of IsiAs has a relatively lower resolution (Supplementary Fig. 3c).

The atomic model of PSI-IsiA was built based on the cryo-EM map (see "Methods"; Supplementary Tables 1–3), which reveals a PSI monomeric core associated with six unique subunits outside of PsaA (Fig. 1a, b). Five of the six outside subunits have six membrane-spanning helices that bind Chls and carotenoids (Cars), which are characteristic of IsiAs[23–25]. The remaining subunit has nine membrane-spanning helices, six of which are similar to the former five subunits. Thus, they were assigned to IsiAs, which were numbered from 1 to 6 clockwisely (Fig. 1a). Among them, two subunits at positions 4 and 5 were assigned to IsiA1, and the subunit at position 1 was assigned to IsiA2; they were named as IsiA1-4, IsiA1-5, and IsiA2-1, respectively. The remaining three subunits IsiA-2, IsiA-3, and IsiA-6 could not be identified as the cryo-EM map in these regions has a lower resolution than that of the overall resolution; therefore, these subunits were modeled as polyalanines (Fig. 1a). This may be due to a higher structural flexibility of IsiA in these regions compared with PSI, and/or partial dissociation of IsiAs from PSI-IsiA, leading to a lower occupancy and hence a lower resolution. The cofactors in the IsiA subunits are summarized in Supplementary Table 3.

### Structure of the PSI monomer
The overall architecture of the PSI-core monomer in the PSI-IsiA supercomplex is similar to that in the PSI tetramer isolated from *Anabaena*[11–13]. The PSI monomer contains 11 subunits, ten of which are PsaA, PsaB, PsaC, PsaD, PsaE, PsaF, PsaI, PsaJ, PsaM, and PsaX (Fig. 1b). The remaining one subunit is located at the position corresponding to PsaK, which was modeled as polyalanines (Supplementary Fig. 4a). This cyanobacterium has the *psaK* gene in addition to two unique genes of *alr5290* and *asr5289* with sequences similar to *psaK*. The amino acid sequences of Alr5290 and Asr5289 have similarities of 40% and 38% with that of PsaK (Supplementary Fig. 4b). However, none of the three sequences can be fitted into the density of the map. This subunit was therefore named Unknown in the present structure. Very interestingly, PsaL is lacking in the PSI-IsiA structure; instead, a C-terminal domain of IsiA2-1 occupies the position of PsaL (see below for details). The PSI core contains 92 Chls *a*, 21 *β*-carotenes, 3 [4Fe-4S] clusters, 2 phylloquinones, and 5 lipid molecules, which are summarized in Supplementary Table 3.

### Structure of IsiA2-1
IsiA2-1 shows an atypical structure among the six subunits of IsiA bound to PSI (Fig. 1). Superposition of the PSI-IsiA structure with the PSI-monomer structure of the *Anabaena* PSI tetramer (PDB: 6JEO) clearly exhibits a structural correspondence between the C-terminal domain of IsiA2-1 and PsaL (Fig. 2a), and the binding sites of Chls and Cars are conserved between the C-terminal domain of IsiA2-1 and PsaL (Fig. 2b). Furthermore, the C-terminal domain of IsiA2 has a high sequence similarity to PsaL (Supplementary Fig. 5); however, the amino acid residues of W426/F427/N451/W454 in IsiA2-1 are remarkably different from the corresponding residues in PsaL (Fig. 2c, d, Supplementary Fig. 5). In addition, the loop region of L325–T343 is clearly connected between the N-terminal and C-terminal domains of IsiA2-1 (Fig. 2e), making it a single polypeptide. These results provide clear evidence for the absence of PsaL in the PSI-IsiA structure observed here. Thus, PsaL is replaced by the C-terminal PsaL-like domain of IsiA2-1 in the *Anabaena* PSI-IsiA supercomplex.

The N-terminal IsiA domain of IsiA2 shows sequence similarity to IsiA1 (65%). The root mean square deviation (RMSD) of the structures between IsiA1–5 and the N-terminal domain of IsiA2-1 is 0.95 Å for 302 Cα atoms (Supplementary Table 5). On the contrary, the C-terminal PsaL-like domain of IsiA2 exhibits sequence similarity to PsaL (62%) and structural similarity with a RMSD value of 0.90 Å for 124 Cα atoms between PsaL and the C-terminal domain of IsiA2-1 (Fig. 2b). IsiA2-1 binds 17 Chls *a* and 5 *β*-carotenes (Fig. 2f, Supplementary Table 3). The axial ligands of the central Mg atom of Chl in IsiA2-1 are provided by main and side chains of amino acids, as well as a water molecule (Supplementary Fig. 6, Supplementary Table 4).

## Structures of IsiA1-4 and IsiA1−5

For the assignments of IsiA1-4 and IsiA1−5, we focused on characteristic amino acid residues among the four types of IsiAs in *Anabaena*. IsiA1 was identified at positions 4 and 5 of the PSI-IsiA supercomplex (Fig. 1a). The amino acid residues of F45/W47/K279/G281/V282/T283 in IsiA1 are different from the corresponding residues in IsiA2, IsiA3, and IsiA5, e.g., IsiA1-F45 vs. IsiA2-M45, IsiA3-T46, IsiA5-T49, etc. (Supplementary Fig. 7a–c). This allows us to assign IsiA1 to positions 4 and 5 in the PSI-IsiA structure. IsiA1-5 binds 17 Chls *a* and 1 *β*-carotene (Fig. 3a). The axial ligands of the central Mg atom of Chl in IsiA1-5 are provided by main and side chains of amino acid residues (Supplementary Fig. 8a, Supplementary Table 4). On the other hand, IsiA1-4 binds only 10 Chls *a* and 1 *β*-carotene (Fig. 3b). The axial ligands of the IsiA1-4 Chls are summarized in Supplementary Fig. 8b and Supplementary Table 4. The Chl content of IsiA1-4 is different from that of IsiA1-5, albeit with the same gene product. This may be due to weaker densities of IsiA1-4 than IsiA1-5, leading to the inability of assignment for some Chl molecules. We determined the present structure of PSI-IsiA according to our criterion in Coot (see Methods). Even though similar local resolutions between the two subunits could be observed (Supplementary Fig. 3c), some of the Chl molecules may be invisible in IsiA1-4. Alternatively, some Chls may be naturally absent in IsiA1-4. This remarkable difference may bring consequences in excitation-energy transfer from IsiAs to the PSI core; however, a solid conclusion has to wait until a higher resolution structure is obtained.

## Structures of IsiA-2, IsiA-3, and IsiA-6

Among the remaining three IsiA subunits (IsiA-2, IsiA-3, and IsiA-6), IsiA-2 has 8 Chls *a*; IsiA-3 has 1 Chl *a* and 1 *β*-carotene; IsiA-6 has 11 Chls *a* (Fig. 4, Supplementary Table 3). Each IsiA subunit shows different pigment compositions, which may partly be due to weak densities in these IsiA subunits (Supplementary Fig. 3c; see Methods), but the

possibility of natural differences among the three IsiAs cannot be excluded. The RMSD values of the structures between IsiA1-5 and IsiA-2/IsiA-3/IsiA1-4/IsiA-6 are 0.56–0.94 Å for a total of 277–326 Cα atoms (Supplementary Table 5).

## Interactions among the IsiA subunits

There are many interactions among the IsiA subunits, which are mostly hydrophobic (Fig. 5a). The amino acid residues A101/A105/L108 and BCR424 in IsiA1-4 interact with Chl molecules of a405/a415 in IsiA1-5 through hydrophobic interactions at distances of 3.3–3.9 Å (the upper panel in Fig. 5b). The amino acid residues A44/F45/W47/F48/S51 and the pigment molecules a404 and BCR424 in IsiA1-4 are also associated with F254/Y258/L278/K279/F280 and a417 in IsiA1-5 through hydrophobic interactions at distances of 3.3–4.7 Å (the lower panel in Fig. 5b). BCR424 in IsiA-3 interacts with L251/F254/V255/Y258 in IsiA1-4 through hydrophobic interactions at distances of 3.5–4.2 Å (Fig. 5c). The amino acid residues L40/A44/F45/A105/L108 in IsiA1-5 are associated with a405/a415/a417 in IsiA-6 through hydrophobic interactions at distances of 3.1–3.8 Å (Fig. 5d). The amino acid residues L40/A44/A105/I108/L113 and BCR524 in IsiA2-1 are coupled with a405/a415/a417 in IsiA-2 through hydrophobic interactions at distances of 3.4–4.0 Å (Fig. 5e). No characteristic interactions between IsiA-2 and IsiA-3 are found in the present structure (black square in Fig. 5a). It should be noted that their interactions remain ambiguous because of weak densities in the corresponding map among IsiAs (Supplementary Fig. 3c).

## Interactions between the IsiA subunits and the PSI core

The interactions between IsiAs and PSI are shown in Fig. 6. The amino acid residues W313/V331/L347 in IsiA2-1 interact with F333 and the Chl molecules a833/a846 in PsaA through hydrophobic interactions at distances of 3.4–3.7 Å, while an oxygen atom of the carbonyl group

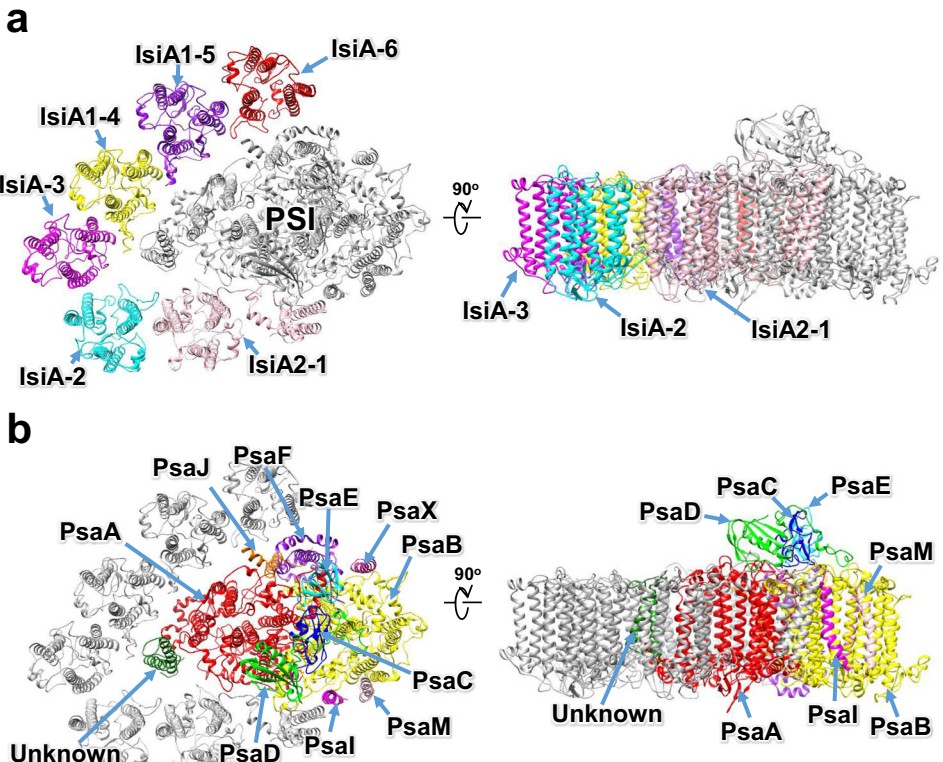

**Fig. 1 | Overall structure of the PSI-IsiA supercomplex from *Anabaena*.** Structures are viewed from the cytosolic side (left panels) and the direction perpendicular to the membrane normal (right panels). Only protein structures are shown, and cofactors are omitted for clarity. The IsiA (**a**) and PSI core (**b**) subunits are labeled with different colors.

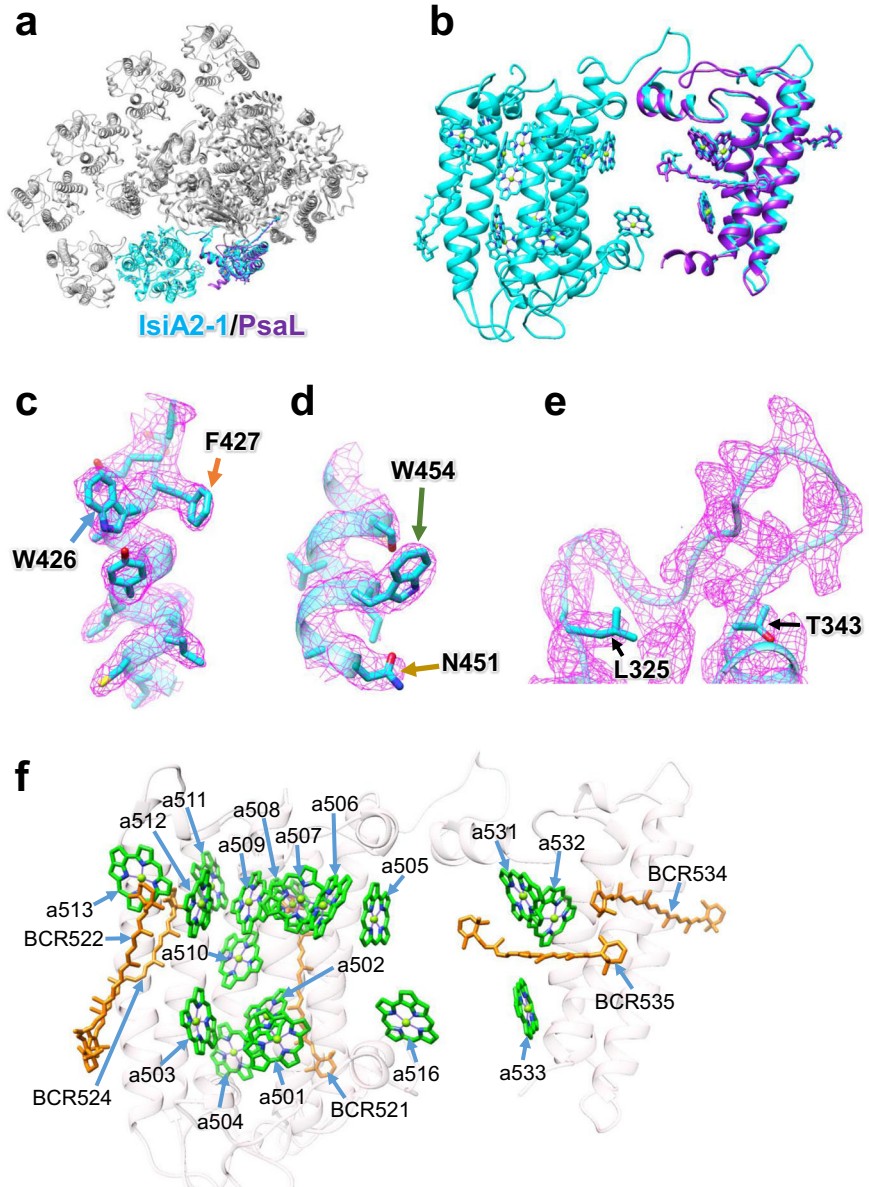

**Fig. 2 | Structure of IsiA2-1. a** Superposition of the PSI-IsiA structure with the PSI-monomer structure prepared from the *Anabaena* PSI tetramer (PDB: 6JEO). The structures are viewed from the cytosolic side. IsiA2-1 and PsaL are colored cyan and purple, respectively, whereas the other subunits are colored gray. **b** Side view of the superposition of structures between IsiA2-1 and PsaL. **c, d** Characteristic maps and residues of the C-terminal PsaL-like domain of IsiA2-1. Residues characteristic of the C-terminal domain of IsiA2-1 are depicted in sticks and labeled. **e** Loop structure of L325–T343 in IsiA2-1. The maps are shown as meshes at 1.5 σ contour level, and the corresponding models are depicted in sticks and ribbons (**c–e**). **f** Structure of IsiA2-1 depicted in ribbons and arrangements of Chl and β-carotene (BCR). Chls and β-carotenes are colored green and orange, respectively. Only rings of the Chl molecules are depicted.

of G341 in IsiA2-1 is hydrogen-bonded with a nitrogen atom of T16 in PsaD at a distance of 3.3 Å (the left panel in Fig. 6b). Moreover, BCR521 in IsiA2-1 is associated with Chl a837 in PsaA at a distance of 3.6 Å (the right panel in Fig. 6b). The amino acid residue I333 in IsiA1-4 interacts with BCR858 in PsaA at a distance of 4.4 Å (Fig. 6c), whereas W14 in IsiA1-4 is located near Chl a821 in PsaA at a distance of 3.5 Å (Fig. 6d). It should be noted that other interactions remain ambiguous because of the local weak densities in the corresponding map among IsiAs and some of the PSI-core subunits (Supplementary Fig. 3c).

## Excitation-energy-transfer processes in the PSI-IsiA supercomplex

Time-resolved fluorescence (TRF) of the PSI-IsiA supercomplex was measured at 77 K and globally analyzed to obtain fluorescence decay-associated (FDA) spectra (Fig. 7). The 55-ps FDA spectrum exhibited positive amplitudes around 685 nm and negative amplitudes around 727 nm. Since a set of positive and negative bands indicates energy transfer from Chl with the positive one to Chl with the negative one, the positive-negative pair of the 685 and 727-nm bands reflects energy transfer from the 685-nm component to the 727-nm component. The 55-ps FDA spectrum also showed a positive shoulder around 694 nm and a negative shoulder around 707 nm. The 120-ps FDA spectrum displayed two positive bands at 690 and 707 nm. This is in striking contrast to the previous results of the *Anabaena* PSI monomer, dimer, and tetramer, in which only a broad band around 728 nm appeared in the 100–170-ps FDA spectra[31]. These results suggest that the *Anabaena* IsiAs affect excitation-energy-transfer processes occurring in the early time region after excitation.

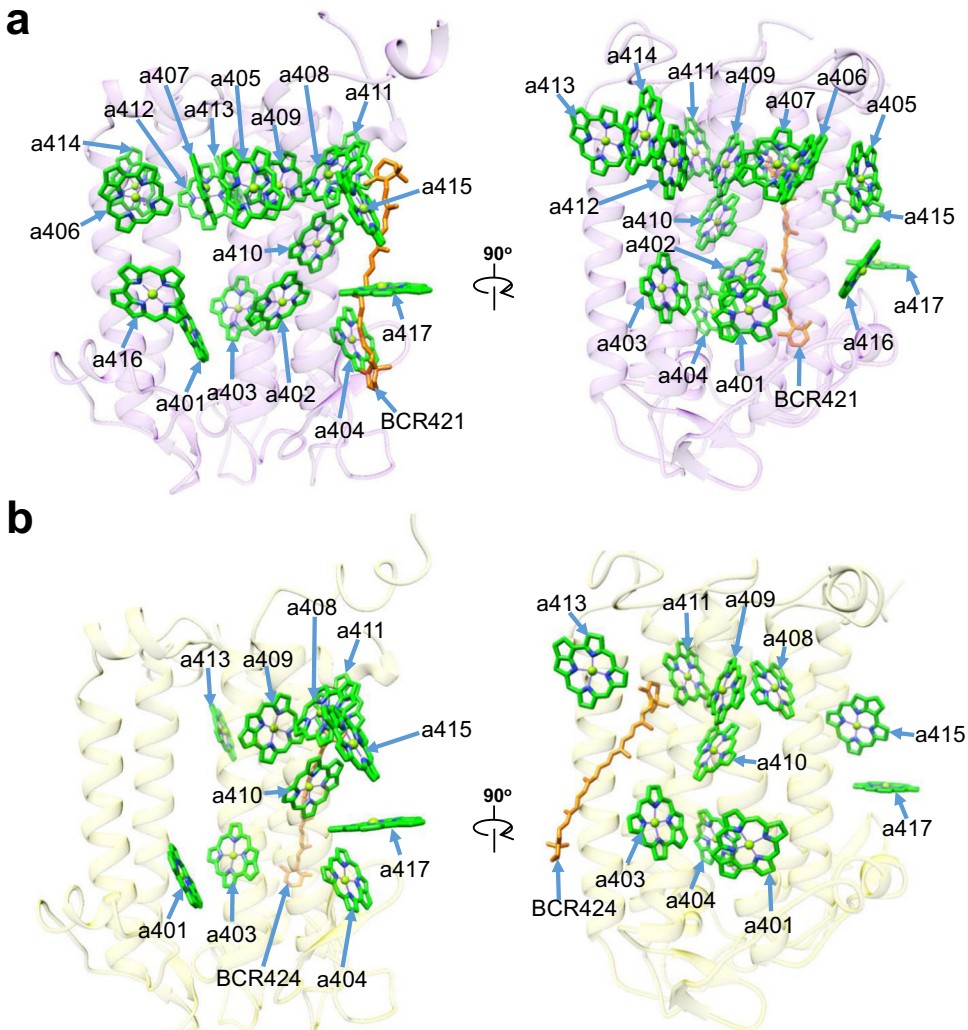

**Fig. 3 | Structures of IsiA1-5 and IsiA1-4.** Structures of IsiA1-5 (**a**) and IsiA1-4 (**b**) depicted in transparent cartoon model and arrangements of Chl and β-carotene (BCR). Chls and β-carotenes are colored green and orange, respectively. Only rings of the Chl molecules are depicted.

## Discussion

This study showed the expression of IsiA1, IsiA2, IsiA3, and IsiA5 at transcript and protein levels (Supplementary Fig. 1a, c), although the expression level of IsiA2 may be somewhat lower than other types of IsiAs. This may reflect a lower number of IsiA2 in the PSI-IsiA structure. Nevertheless, the expression of all *isiA* genes in *Anabaena* is in good agreement with the results of *Leptolyngbya* showing that the five types of IsiA proteins were biosynthesized under the iron-deficient condition[29]. Thus, cyanobacteria with more than one *isiA* gene may accumulate all *isiA* products under iron-deficient conditions.

The structure of the PSI-IsiA supercomplex reveals the binding of six IsiA subunits to the PSI-monomer core, three of which were identified as IsiA2-1, IsiA1-4, and IsiA1-5, whereas the remaining three subunits (IsiA-2, IsiA-3, and IsiA-6) could not be identified (Fig. 1a). The structures of the five IsiA subunits other than IsiA2-1 are similar to that of the structurally known IsiA subunit of other cyanobacteria, with the characteristic six trans-membrane helices[23–25]. This is in good agreement with the result of the sequence alignment that puts the *Anabaena* IsiA1 into the same group of the structurally known IsiA family[30]. The *Anabaena* IsiA1 has high sequence and structural similarities with the *Anabaena* IsiA2, IsiA3, and IsiA5 (Supplementary Fig. 7a), with RMSDs of the strctures between IsiA1-5 and IsiA2-1/IsiA-2/IsiA-3/IsiA1-4/IsiA-6 in the range of 0.56–0.95 Å (Supplementary Table 5). Because all four IsiAs in *Anabaena* were expressed under

our experimental conditions (Supplementary Fig. 1a, c), these findings indicate that some of the unidentified subunits of IsiA-2, IsiA-3, and IsiA-6 are at least the gene products of *isiA3* and *isiA5* in the PSI-IsiA supercomplex. However, based on the lower expression level, *isiA2* may be excluded among the unidentified IsiAs at positions 2, 3 and 6.

Among the six IsiA subunits, IsiA2-1 shows an unusual structure different from the other five IsiA subunits, because it has an extra C-terminal PsaL-like domain (Figs. 2a, 2b, Supplementary Fig. 5). Bryant and co-workers reported that IsiA4 of *Leptolyngbya* contained a PsaL-like domain[29] similar to the sequence of the *Anabaena* IsiA2[30]. The authors also suggested that the *Leptolyngbya* IsiA4 was related to protein aggregation and the formation of supercomplexes with PSI[29]. The present structure indicates that cyanobacteria having IsiAs with a PsaL-like domain may organize PSI-monomer-IsiA supercomplexes through the replacement of PsaL by IsiAs containing a PsaL-like domain under iron-deficient conditions.

It is interesting to note that the *Anabaena* PsaL is still expressed under the iron-deficient condition and then contributes to the formation of PSI tetramers, as PSI tetramers were detected by two-dimensional BN/SDS-PAGE using the iron-limited thylakoids[30]. Since PsaL plays a crucial role in the oligomerization of PSI[32–34], it seems that PsaL competes with IsiA2 for interactions with PSI, resulting in the formation of a PSI monomer which contains IsiA2 without PsaL or a PSI

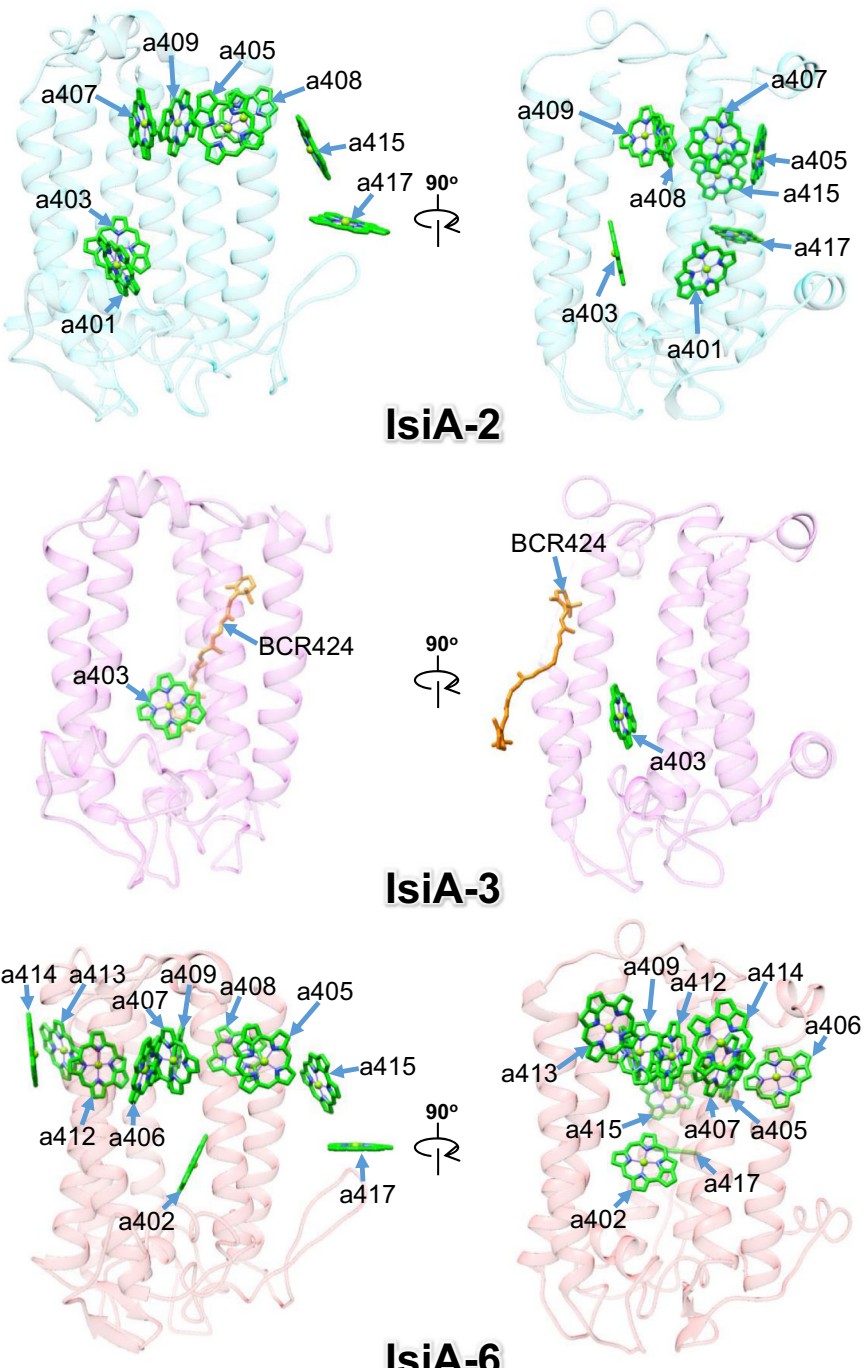

**Fig. 4 | Structures of IsiA-2, IsiA-3, and IsiA-6.** Structures of IsiA-2, IsiA-3, and IsiA-6 depicted in transparent cartoon model and arrangements of Chl and β-carotene (BCR). Chls and β-carotene are colored green and orange, respectively. Only rings of the Chl molecules are depicted.

tetramer which contains PsaL but without IsiA2, under iron-deficient conditions.

Based on the observations obtained in the present study, we propose a schematic model for the assembly of the PSI-monomer-IsiA supercomplex and PSI tetramer in *Anabaena* under iron-deficient conditions (Fig. 8). The PsaL and IsiA2 subunits play important roles in determining the oligomeric states of PSI in *Anabaena*. Once IsiA2 is bound to a PSI monomer without PsaL, oligomerization of monomeric PSI cores to dimers and tetramers is inhibited by the N-terminal IsiA domain of IsiA2, which is located near PsaA (Fig. 1). The PSI-IsiA2 supercomplex subsequently assembles into a PSI-monomer-IsiA supercomplex with the association of the remaining five IsiA

subunits. As a result, no PSI-tetramer-IsiA fraction was obtained, as observed in the two-dimensional BN/SDS-PAGE[30]. In contrast, when PsaL is first bound to a PSI monomer without IsiA2, further assembly to PSI tetramers may proceed, which exclude the binding of IsiA2. The competitive assembly of oligomeric PSI cores and the existence of a PSI-monomer-IsiA supercomplex would often occur in cyanobacteria expressing both PsaL and IsiAs having a PsaL-like domain in iron-limited environments.

Indeed atomic force microscopy of native thylakoids from *S. elongatus* PCC 7942 showed various types of PSI-IsiA supercomplexes[35], although the cryo-EM single-particle analysis of PSI-IsiA isolated from this cyanobacterium displayed a typical

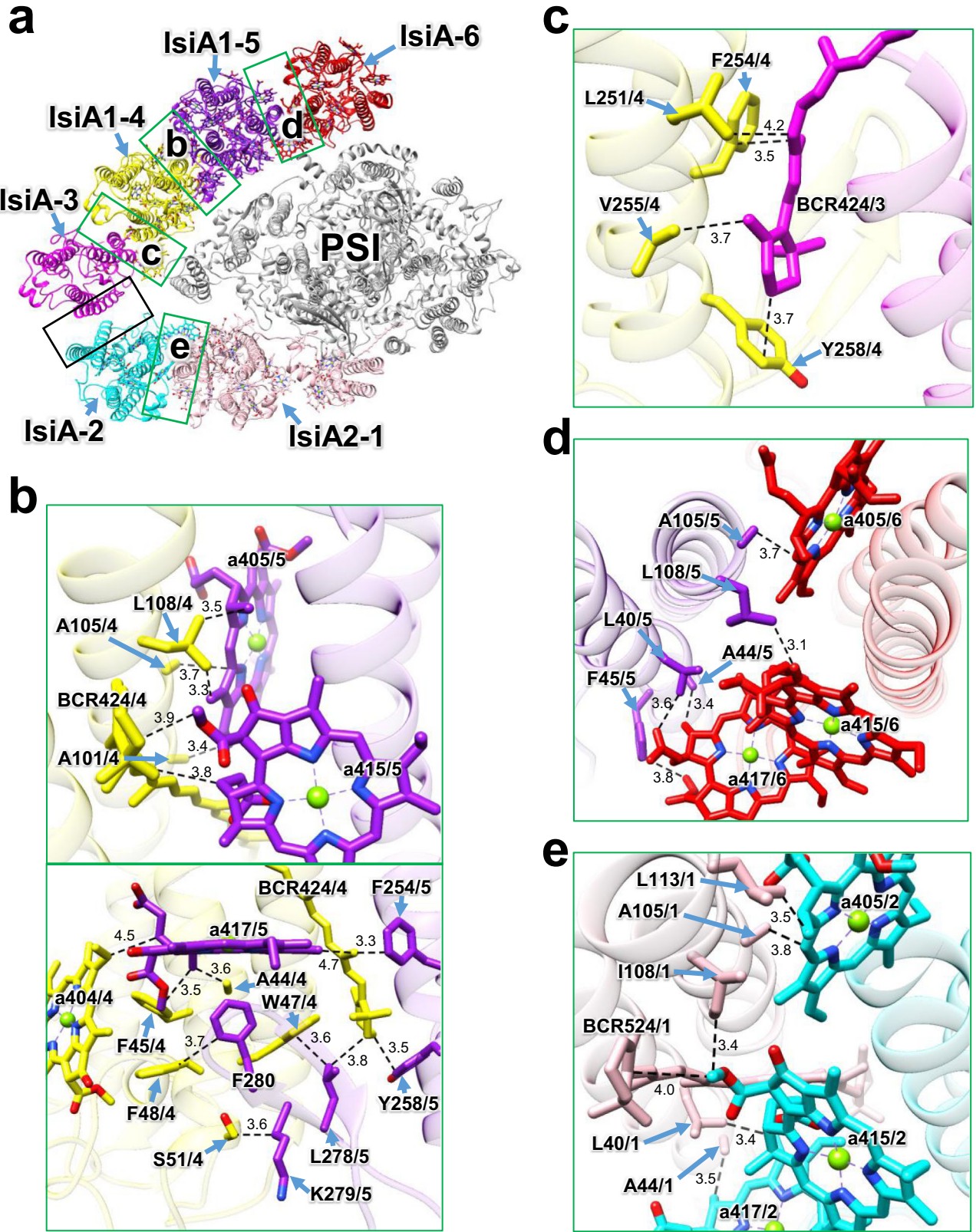

**Fig. 5 | Interactions among the IsiA subunits. a** Overall structure of PSI-IsiA viewed from the cytosolic side. The green squared areas are enlarged in panels **b**–**e**, whereas the black squared area does not have a characteristic interaction in the present structure. **b** Interactions between IsiA1-4 and IsiA1-5 (upper and lower panels). **c**, Interactions between IsiA-3 and IsiA1-4. **d** Interactions between IsiA1-5 and IsiA-6. **e** Interactions between IsiA2-1 and IsiA-2. Interactions are indicated by dashed lines, and the numbers are distances in Å. Amino acid residues and pigments participating in the interactions are labeled; for example, A105/4 means Ala105 in IsiA1-4; a415/5 means Chl *a* 415 in IsiA1-5. BCR, *β*-carotene.

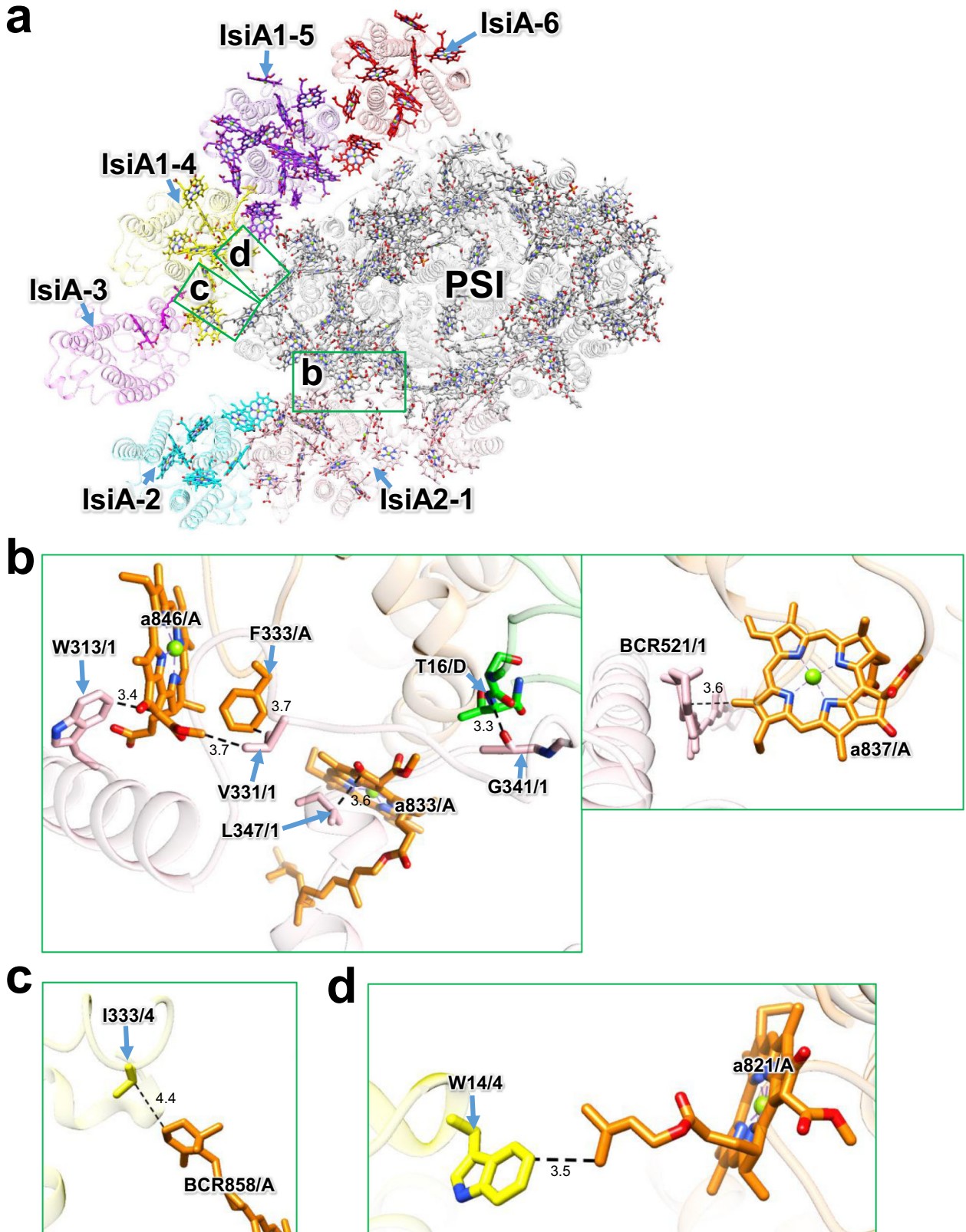

**Fig. 6 | Interactions between IsiAs and PSI. a** Overall structure of PSI-IsiA viewed from the cytosolic side. Green squared areas are enlarged in panels **b**–**d**. **b** Interactions between IsiA2-1 and PsaA/PsaD (left and right panels). **c** Interaction between IsiA1-4 and PsaA. **d** Interaction between IsiA1-4 and PsaA.

Interactions are indicated by dashed lines, and the numbers are distances in Å. Amino acid residues and pigments participating in the interactions are labeled; for example, W313/1 means Trp313 in IsiA2-1; a846/A means Chl *a* 846 in PsaA. BCR, *β*-carotene.

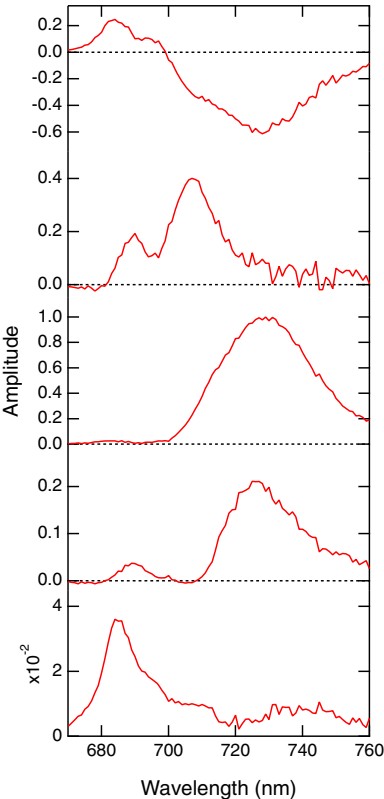

**Fig. 7 | Fluorescence decay-associated spectra of PSI-IsiA.** TRF was measured at 77 K upon excitation at 445 nm, followed by global analysis to construct the FDA spectra. The analyzed time constants for the spectra were 55 ps, 120 ps, 520 ps, 1.2 ns, and 3.9 ns from top to bottom. The FDA spectra were normalized by the maximum amplitude of the 520-ps spectrum.

structure of PS-IsiA, namely a PSI trimer with a closed ring of 18 IsiAs[24]. These observations imply that the majority of PSI-IsiA in the native thylakoids of *S. elongatus* PCC 7942 is a PSI-trimer-IsiA-ring supercomplex, but there may be other minor supercomplexes in the membrane. The present PSI-monomer-IsiA structure may represent one of the major forms of PSI-IsiA supercomplexes in native thylakoids from *Anabaena*.

The binding of IsiAs to PSI affected the absorption and fluorescence spectra (Supplementary Fig. 1d, e). The absorption spectrum of PSI-IsiA exhibited the Qy band of Chl *a* at 673 nm (Supplementary Fig. 1d), which was shorter than that in the spectrum of the PSI monomer from *Anabaena*[11]. This is characteristic of the existence of IsiAs, as observed in the PSI-IsiA supercomplexes from various cyanobacteria[21,26,28]. Such a blue-shifted band has also been observed in the absorption spectrum of thylakoid membranes prepared from *Anabaena* grown under the iron-deficient condition[30]. Furthermore, the fluorescence-emission spectrum of PSI-IsiA showed two bands at 687 and 727 nm (Supplementary Fig. 1e). The former appears to originate from IsiAs as observed in the *Anabaena* thylakoids[30] and other cyanobacteria[21,36,37], whereas the latter originates from one of the specific low-energy Chls in PSI, which has been denoted as Low2 based on its absence in *Gloeobacter violaceus* PCC 7421 and *Synechocystis* sp. PCC 6803[38]. In contrast, the fluorescence spectrum of the PSI monomer of *Anabaena* grown under iron-replete conditions has shown a band at 730 nm, without the characteristic 687-nm band observed in the PSI-IsiA spectrum[11,31]. Thus, the spectroscopic properties of the *Anabaena* IsiAs clearly appeared even under steady-state experimental conditions.

The excitation-energy dynamics in the *Anabaena* PSI-IsiA supercomplex have been examined by the FDA spectra (Fig. 7). The

characteristic fluorescence band at 684 nm originating from IsiAs decayed with a time constant of 55 ps, and the corresponding rise of fluorescence appeared around 727 nm, suggesting excitation-energy transfer from IsiAs to PSI. In addition to the negative 727-nm band, a negative shoulder was recognized around 707 nm in the 55-ps FDA spectrum, which was followed by a positive 707-nm band in the 120-ps FDA spectrum. Since the distinct positive band at 707 nm was not observed in the *Anabaena* PSI monomer without IsiAs[31], it is suggested that this clear 707-nm band may occur by interactions between IsiAs and PSI in *Anabaena*. In contrast to the 55-ps FDA spectrum exhibiting the positive-negative pair, the 690-nm band lacking a corresponding negative band in the 120-ps FDA spectrum may be mainly attributed to excitation-energy quenching through interactions among pigments around/within IsiAs. Energy quenching with time constants of hundreds of picoseconds has been interpreted by various spectroscopic studies using photosynthetic pigments and LHCs[39,40]. Thus, the *Anabaena* PSI-IsiA may possess quenching sites at 690 and 707 nm, the latter having the same transition energy as found in the *Anabaena* PSI tetramer[11,31]. A 686-nm band appeared in the 3.9-ns FDA spectrum, suggesting that uncoupled Chls within IsiAs cannot transfer excitation energy to other pigments, which was similar to that observed in the *T. vulcanus* PSI-IsiA[25].

Based on the properties of excitation-energy-transfer processes and the structure of the *Anabaena* PSI-IsiA, we propose excitation-energy-transfer pathways from IsiAs to PSI. The excitation energy in IsiA2-1, IsiA1-4, and IsiA-6 appear to be directly transferred to the PSI core, due to the close pigment-pigment interactions of these IsiA subunits with PSI subunits (Supplementary Fig. 9a). On the contrary, the energy in IsiA-2, IsiA-3, and IsiA1-5 may be transferred once to the neighboring IsiA subunits prior to excitation-energy transfer to PSI. As excitation-energy transfer from IsiAs to PSI occurred with a time constant of 55 ps (Fig. 7), the Chl couplings of IsiA2-1-a516/IsiA2-1-a533, IsiA2-1-a508/PsaA-a846, IsiA1-4-a415/PsaA-a845, IsiA1-4-a417/PsaA-a845, IsiA1-4-a404/PsaA-a816, and IsiA-6-a417/PsaK-a101 (Supplementary Fig. 9b–f) may be good energy donors and acceptors between IsiAs and PSI. As for IsiA2-1, the pigment molecules in the C-terminal PsaL-like domain may function similarly to those in PsaL, because of the almost same pigment arrangements between the C-terminal domain of IsiA2-1 and PsaL (Fig. 2b). Therefore, the Chl couplings between the N-terminal and C-terminal domains of IsiA2-1 may be involved in excitation-energy transfer with a time constant of 55 ps. Furthermore, we propose that the Car-Chl coupling of IsiA2-1-BCR521/PsaA-a837 (Supplementary Fig. 9c) may contribute to either ultrafast excitation-energy transfer in the time order of femtoseconds as observed in ultrafast spectroscopies[39] or energy quenching with a time constant of 120 ps by Chl-Car interactions between IsiA2-1 and PsaA.

In the PSI-trimer-IsiA structures from other cyanobacteria, six IsiA subunits were associated with a PSI-monomer core[23–25]. Here, we compare the binding properties of IsiAs between *Anabaena* and other cyanobacteria (Supplementary Fig. 10). The PSI-monomer cores are well fitted between *Anabaena* and *Synechocystis* sp. PCC 6803 (Supplementary Fig. 10a). The six IsiA subunits in the *Synechocystis* PSI-IsiA were named IsiA-1 to IsiA-6 (Supplementary Fig. 10a). The IsiA1-4, IsiA1-5, and IsiA-6 subunits in the *Anabaena* PSI-IsiA are located at similar positions as the IsiA-1, IsiA-2, and IsiA-3 subunits, respectively, in the *Synechocystis* PSI-IsiA; these three IsiAs are associated with PsaA in the two species (Supplementary Fig. 10a). However, the remaining three IsiA subunits are bound to the outside of PsaB in the *Synechocystis* PSI-IsiA, whereas they are bound to the remaining part of PsaA opposite to the side of PsaB in the *Anabaena* PSI-IsiA. The locations of IsiA to the outside of PsaB in the other species may allow the formation of a complete ring surrounding a trimeric PSI core, whereas positions 1–3 in *Anabaena* would clash with oligomeric PSIs. The different binding pattern between *Anabaena* and other cyanobacteria may be due to differences in the structures of the IsiA1-4, IsiA1-5, and IsiA-6 subunits

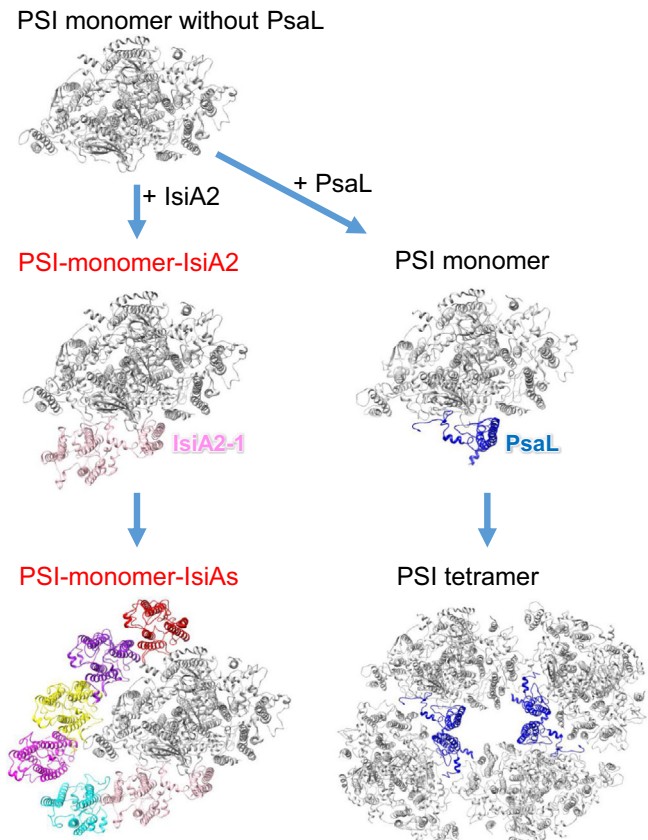

**Fig. 8 | An assembly model proposed for the *Anabaena* PSI-IsiA supercomplex.** The structures are viewed from the cytosolic side. The PSI monomer and tetramer structures are taken from 6JEO (PDB ID).

in the *Anabaena* PSI-IsiA compared with the corresponding IsiA subunits in the *Synechocystis* PSI-IsiA (Supplementary Fig. 10b). In particular, the structural difference between IsiA-6 of *Anabaena* and IsiA-3 of *Synechocystis* is much larger than that between IsiA1-4/IsiA1-5 of *Anabaena* and IsiA-1/IsiA-2 of *Synechocystis* (Supplementary Fig. 10b). The structural distortion of the *Anabaena* IsiAs may also occur by the unusual binding of IsiA2-1, IsiA-2, and IsiA-3 to PSI. Moreover, the C-terminal loop region observed in the *Synechocystis* IsiAs is missing in the IsiA1-4, IsiA1-5, and IsiA-6 subunits in the *Anabaena* PSI-IsiA structure (yellow arrows in Supplementary Fig. 10b). These structural differences are also observed between *Anabaena* and *T. vulcanus* NIES-2134 (Supplementary Fig. 10c) and between *Anabaena* and *S. elongatus* PCC 7942 (Supplementary Fig. 10d), which may lead to the suppression of the binding of IsiAs to the outside of PsaB in *Anabaena*.

Our previous sequence analysis has shown that the putative amino acid ligands of Chls in the *Anabaena* IsiA1 are consistent with the Chl ligands in the structurally known IsiAs from the three types of cyanobacteria[30]. Here, we show a different amino acid ligand between Q34 of IsiA1-5 and H34 of IsiA2-1 (Supplementary Figs. 6, 7a, 8a). In addition, two Chls corresponding to a414 and a415 in IsiA1-5 are lacking in IsiA2-1, because IsiA2−1 has I144 and S309 instead of H144 and Q316 of IsiA1−5 (Supplementary Figs. 6, 7a, 8a). The variations in the amino acid ligands of Chls in the *Anabaena* IsiAs may be important for light-harvesting strategy in cyanobacteria having several *isiA* genes.

Another difference is the structure of PsaK between *Anabaena* and other cyanobacteria. In the PSI-monomer-IsiA structure of *Anabaena*, there is a polypeptide modeled as Unknown in the position of PsaK, as we were unable to model it based on the sequence of PsaK. Although there are two other genes (*alr5290* and *asr5289*) in *Anabaena* that have sequence similarities with PsaK, it is unclear which gene product occupies the PsaK position. This may be another distinct feature for the structure of the PSI-monomer-IsiA in comparison with the PSI-IsiA structures from other cyanobacteria.

In conclusion, this study demonstrates the overall structure of a PSI-monomer-IsiA supercomplex from *Anabaena* grown under the iron-deficient condition. The structure of IsiA2-1 shows the N-terminal IsiA and C-terminal PsaL-like domains, and is bound to PSI by substituting PsaL with the C-terminal PsaL-like domain. The binding of IsiA2 to PSI in *Anabaena* leads to an inhibition of PSI oligomerization, resulting in a monomeric PSI core with six IsiA subunits bound. Unlike the typical PSI-trimer-IsiA structures, the IsiA subunits are associated with the PsaA side but not with the outside of PsaB. This may cause differences in interactions among IsiAs and between IsiAs and PSI core, between *Anabaena* and other cyanobacteria having PSI-trimer-IsiA supercomplexes. These structural findings may characterize cyanobacteria with multiple copies of *isiA* genes, which may provide a survival strategy for such cyanobacteria under iron-limited conditions.

## Methods

### Cell growth and preparation of thylakoid membranes

The *Anabaena* cells were grown in an iron-replete BG11 medium supplemented with 10 mM HEPES-KOH (pH 8.0) at a photosynthetic photon flux density of 30 µmol photons $m^{-2}$ $s^{-1}$ at 30 °C with bubbling of air containing 3% (v/v) $CO_2$[30]. For iron starvation, the cells grown under the iron-replete condition were substituted with an iron-free BG11 medium and then cultured according to the method of Nagao et al.[30]. The cells were harvested by centrifugation, and then thylakoid membranes were prepared by agitation with glass beads[41], followed by suspension with a 20 mM MES-NaOH (pH 6.5) buffer containing 0.2 M trehalose, 5 mM $CaCl_2$, and 10 mM $MgCl_2$ (buffer A).

### Transcription analysis of IsiAs

The cells were grown for 0 and 20 days under the iron-deficient condition, and were collected by centrifugation at 4 °C and stored at −80 °C until use. Total RNA was extracted from the cells with a PGTX solution[42] and purified using NucleoSpin RNA (MACHEREY-NAGEL). cDNA was synthesized from 1 µg of total RNA using ReverTra Ace qPCR RT Master Mix (TOYOBO), including random and oligo (dT) primers. The presence of contaminating genome DNA was confirmed by incubating 1 µg of total RNA without reverse transcriptase under the condition of cDNA synthesis. PCR was conducted with cDNA as the template using Tks Gflex DNA Polymerase (Takara Bio). Quantitative reverse transcription PCR (qRT-PCR) was performed according to the method of Ehira and Miyazaki[43] using Go Taq qPCR Master Mix (Promega). Primers used for PCR and qRT-PCR are listed in Supplementary Table 6.

### Purification of the PSI-IsiA supercomplex

Thylakoid membranes were solubilized with 1% (w/v) *n*-dodecyl-β-D-maltoside (β-DDM) at a Chl concentration of 0.25 mg $mL^{-1}$ for 10 min on ice in the dark with gentle stirring. After centrifugation at 100,000 × *g* for 10 min at 4 °C, the resultant supernatant was loaded onto a Q-Sepharose anion-exchange column (1.6 cm of inner diameter and 25 cm of length) equilibrated with buffer A containing 0.03% β-DDM (buffer B). The column was washed with buffer B containing 50 mM NaCl until the eluate became colorless. Two types of buffers, buffer B and buffer C (buffer B containing 500 mM NaCl), were used for the elution of PSI-IsiA from the column in a linear-gradient step of 0–600 min, 10–50% buffer C at a flow rate of 2.0 mL $min^{-1}$. The PSI-IsiA-enriched fraction was eluted at around 200–230 mM NaCl.

The eluted PSI-IsiA were precipitated by centrifugation after the addition of polyethylene glycol 1500 to a final concentration of 15% (w/v), and then suspended with Buffer B. The resultant PSI-IsiA samples were loaded onto a linear gradient containing 10–40% (w/v) trehalose

in a medium of 20 mM MES-NaOH (pH 6.5), 5 mM $CaCl_2$, 10 mM $MgCl_2$, 100 mM NaCl, and 0.03% $\beta$-DDM. After centrifugation at 154,000 × $g$ for 18 h at 4 °C (P40ST rotor; Hitachi), a major green fraction (Supplementary Fig. 1b) was collected and concentrated using a 150 kDa cut-off filter (Apollo; Orbital Biosciences) at 4000 × $g$. The concentrated samples were stored in liquid nitrogen until use.

### Biochemical and spectroscopic analyses of the PSI-IsiA supercomplex

Subunit composition of the PSI-IsiA supercomplex was analyzed by SDS-polyacrylamide gel electrophoresis (PAGE) containing 16% (w/v) acrylamide and 7.5 M urea according to the method of Ikeuchi and Inoue[44] (Supplementary Fig. 1c). The PSI-IsiA supercomplexes (4 μg of Chl) were solubilized by 3% lithium lauryl sulfate and 75 mM dithiothreitol for 10 min at 60 °C, and loaded onto the gel. A standard molecular weight marker (SP-0110; APRO Science) was used. The subunit bands were assigned by mass spectrometry according to the method of Nagao et al.[45]. Pigment composition of the PSI-IsiA supercomplex was analyzed by HPLC according to the method of Nagao et al.[46], and the elution profile was monitored at 440 nm (Supplementary Fig. 1 f).

Absorption spectrum was measured under room-temperature conditions using a UV–Vis spectrophotometer (UV-2450; Shimadzu) (Supplementary Fig. 1d). Steady-state fluorescence-emission spectrum was measured at 77 K using a spectrofluorometer (RF-5300PC; Shimadzu) (Supplementary Fig. 1e). TRF was measured at 77 K and recorded three times by a time-correlated single-photon counting system with a wavelength interval of 1 nm and a time interval of 2.44 ps[47]. A picosecond pulse diode laser (PiL044X; Advanced Laser Diode Systems) was used as an excitation source, and was operated at 445 nm with a repetition rate of 3 MHz. The TRF-measurement conditions were described in detail[48]. The fluorescence intensities were obtained as a function of time ($t$) and wavelength ($\lambda$), and globally analyzed with time constants ($\tau$) independent of $\lambda$, as $\sum_{i=1}^{5} A_i(\lambda) \exp(-t/\tau_i)$. Here, $A_i(\lambda)$ is the FDA spectrum with $\tau_i$. The weighted residual map between the measured and calculated data is shown in Supplementary Fig. 11.

### Cryo-EM data collection

For cryo-EM experiments, 3-μL aliquots of the *Anabaena* PSI-IsiA supercomplex (0.53 mg Chl mL$^{-1}$) in a 20 mM MES-NaOH (pH 6.5) buffer containing 5 mM $CaCl_2$, 10 mM $MgCl_2$, 100 mM NaCl, and 0.03% $\beta$-DDM were applied to Quantifoil R0.6/1, Cu 200 mesh grids pretreated by gold sputtering. Without waiting for incubation, excess solutions were blotted off for 6 sec with a filter paper in the chamber of FEI Vitrobot Mark IV at 4 °C under 100% humidity. The grids were plunged into liquid ethane cooled by liquid nitrogen and then transferred into a CRYO ARM 300 electron microscope (JEOL) equipped with a cold-field emission gun operated at 300 kV. AI detection of hole positions was carried out with yoneoLocr[49], preventing stage alignment failure. All image stacks acquired were collected from 5 × 5 holes per stage adjustment to the central hole using SerialEM[50] and JAFIS Tool version 1 (developed by Dr. Bartosz Marzec, JEOL), which synchronized image shifts with beam tilts, objective stigmas for removal of axial coma aberrations, and two-fold astigmatism. The images were zero-loss energy filtered and recorded at a nominal magnification of × 100,000 on a direct electron detection camera (Gatan K3, AMETEK) with a nominal defocus range of −1.8 to −0.8 μm. One-pixel size corresponded to 0.495 Å. Each image stack was exposed at a dose rate of 13.6 e⁻ Å$^{-2}$ s$^{-1}$ for 3.0 s in CDS mode, and consisted of dose-fractionated 50 movie frames. In total 22,575 image stacks were collected. It is known that photosystem-LHC supercomplexes are easily dissociated during preparations of samples and/or cryo-EM grids. Therefore, acquiring a large amount of image data is often required for 3D reconstruction of supercomplex structures.

### Cryo-EM image processing

The resultant movie frames were aligned and summed using MotionCor2[51] to yield dose-weighted images. Estimation of the contrast transfer function (CTF) was performed using CTFFIND4[52]. All of the following processes were performed using RELION3.1[53]. In total 1,775,806 particles were automatically picked up and used for reference-free 2D classification. Then, 280,489 particles were selected from good 2D classes and subsequently subjected to 3D classification without any symmetry. An initial model for the first 3D classification was generated de novo from 2D classification. As shown in Supplementary Fig. 2c, the final PSI-IsiA structure was reconstructed from 47,602 particles. The overall resolution of the cryo-EM map was estimated to be 2.62 Å by the gold-standard FSC curve with a cut-off value of 0.143 (Supplementary Fig. 3a)[54]. Local resolutions were calculated using RELION (Supplementary Fig. 3c).

### Model building and refinement

Two types of the cryo-EM maps were used for the model building of the PSI-IsiA supercomplex: one was a postprocessed map, and the other was a denoised map using Topaz version 0.2.4[55]. The postprocessed map was denoised using the trained model in 100 epochs with two half-maps. In particular, while most of the subunits could be traced according to their sequences, the three subunits of IsiA-2, IsiA-3, and IsiA-6 were modeled with polyalanines using the denoised map (Supplementary Fig. 12). Each subunit of the homology models constructed using the Phyre2 server[56] was first manually fitted into the two maps using UCSF Chimera[57], and then their structures were inspected and manually adjusted against the maps with Coot[58]. Each model was built based on interpretable features from the density maps with the contour levels of 1.0 and 2.5 σ in the denoised and postprocessed maps, respectively. The complete PSI-IsiA structure was refined with phenix.real_space_refine[59] and Servalcat[60] with geometric restraints for the protein-cofactor coordination. The final model was validated with MolProbity[61], EMRinger[62], and $Q$-score[63]. The statistics for all data collection and structure refinement are summarized in Supplementary Tables 1, 2. All structural figures were made by PyMOL[64] and UCSF Chimera.

Since the numbering of Chls and Cars in this paper were different from those of the PDB data, we listed the relationship of the pigment numbering in this paper with those in the PDB data in Supplementary Table 7.

### Reporting summary

Further information on research design is available in the Nature Portfolio Reporting Summary linked to this article.

## Data availability

Atomic coordinate and cryo-EM maps for the reported structure have been deposited in the Protein Data Bank under an accession code 7Y3F and in the Electron Microscopy Data Bank under an accession code EMD-33593. Source data are provided with this paper.

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

## Acknowledgements

We thank Ms. Kumiyo Kato for her assistance in this study. This work was supported by JSPS KAKENHI grant Nos. JP20H02914 (Koji.K.), JP21K19085 (R.N.), JP20K06528 (Keisuke.K.), and JP17H06434 and JP22H04916 (J.-R.S.), JST-Mirai Program Grant Number JPMJMI20G5 (K.Y.), Takeda Science Foundation (Koji.K.), and the Cyclic Innovation for Clinical Empowerment (CiCLE) from the Japan Agency for Medical Research and Development, AMED (T.H., Keisuke.K., K.Y.).

## Author contributions

R.N. conceived the project; R.N., N.T., and S.S. prepared the PSI-IsiA supercomplex and analyzed its biochemical characterization; S.E. performed transcription analysis of IsiAs; Y.U., M.F., and S.A. performed TRF measurements of the PSI-IsiA supercomplex and their data analysis; T.S. and N.D. identified subunits in the PSI-IsiA supercomplex; T.H. collected cryo-EM images; Koji.K. processed the cryo-EM data and reconstructed the final cryo-EM map; Koji.K. built the structural model and refined the final model; Y.N. analyzed structural data; Keisuke.K. commented on the structural data; K.Y. and J.-R.S. supervised this project; R.N. and S.A. wrote the draft manuscript; and R.N., S.A., and J.-R.S. wrote the final manuscript, and all of the authors joined the discussion of the results.

## Competing interests

The authors declare no competing interests.
