## [Peer Review File · Nature Communications]

Structure of a monomeric photosystem I core associated with iron-stress-induced-A proteins from *Anabaena* sp. PCC 7120REVIEWER COMMENTS

Reviewer #1 (Remarks to the Author):

The manuscript by Nagao et al. describes high-resolution structures of photosystem I from the cyanobacterium *Anabaena* grown under iron-deficient condition. The structure of PSI-IsiA supercomplex was solved by cryo-electron microscopy (cryo-EM) single-particle analysis. The results showed that PSI-IsiA supercomplex consists of six IsiA subunits encoded by different *isiA* genes associated at one site of a monomeric PSI core, forming an unclosed, monomeric PSI-IsiA supercomplex. This is a novel finding showing once again the plasticity of PSI in various organisms responding to environmental stress.

The manuscript is very interesting and accomplished by highly capable scientists. For the benefit of the work and the readers it could be shortened. I would recommend accepting it for publication after addressing some obstacles.

The FSC curves of PSI-IsiA presented in Supplementary Fig. 3a. is not of a high quality we used to see from the corresponding author laboratory. This indicates some local structural problems. Indeed, throughout the manuscript it is stated that interactions could not be concluded because of poor local density. The interactions of chlorophylls with ligands are discussed in detail and I would recommend shorten those descriptions that have no bearing on the mechanism. The relevant PDBs will be published and the readers that are interested in specific aspects will be able to obtain the information. A previous study from the same group using two-dimensional BN/SDS-PAGE analysis of thylakoids from *Anabaena* grown under the iron-deficient condition detected only IsiA1 subunit in the fraction of the PSI-IsiA supercomplex (Nagao, R. et al. 2021). This is a prime example for the fact that blue-native gels produce more artifacts than facts. The explanation for this discrepancy is not convincing. Accordingly, remove panel d from supplementary Figure 1. and minimize the conclusion drawn throughout the manuscript relying on this technique.

The discovery of C-terminal PsaL-like domain in IsiA is highly interesting and merits publication of this work.

Minor corrections

82 related to *isiA1* of *Anabaena30*. Immunological analysis of IsiAs detected one band in Immunological analysis of IsiAs detected identified one band in thylakoids of *Anabaena* prepared from the iron-deficient cells.

Please clarify

84 In addition, two-dimensional blue-native (BN)/SDS-PAGE analysis together with mass spectrometry detected the *IsiA1* subunit in a PSI fraction located near the band of the PSII dimer, showing the formation of a PSI-*IsiA1* supercomplex in *Anabaena30*

Correct

152 the substitution of PsaK with Unknown or inhibit the expression of PsaK.

Reviewer #2 (Remarks to the Author):

This submission presents surprising and interesting discoveries about the structural interactions in a complex of photosystem I with iron-stress-induced protein A (PSI-IsiA) in the filamentous nitrogen-fixing cyanobacterium *Anabaena* sp. PCC 7120. As Fe is an essential nutrient, photosynthetic organisms and cyanobacteria in particular have evolved various mechanisms for Fe homeostasis and acclimation to Fe-limiting conditions. One mechanism among cyanobacteria is the synthesis of IsiA proteins that are homologous to the PSII core antenna protein PsaC. IsiA are known to form various ring-like structures around PSI, and to enhance the PSI absorption cross section, which compensates for the reduced synthesis of phycobiliproteins and PSI under Fe limitation. This contribution reveals the first medium-resolution structure of PSI-IsiA from *Anabaena* and one of a very few structures available overall – the others being from *Thermotrichus vulcanus* published in 2020 by the same

group and from *Synechocystis* sp. (Toporik et al. 2019). Both of these structures contain identical copies of IsiA forming a ring around a PSI trimer, whereas the *Anabaena* complex is rather special, containing different IsiA subunits occupying different sites around a monomeric PSI complex. These studies bring into light an important adaptation mechanism and design principles by which cyanobacteria regulate the antenna size of the photosynthetic reaction centres.

The present structure is probably not the definitive one, as not all subunits and cofactors can be identified due to insufficient resolution. Nonetheless, the authors reveal some novel aspects deserving attention. They show that multiple *isiA* genes are expressed in *Anabaena* under iron-limiting conditions and the respective IsiA proteins occupy specific binding sites at the periphery of the PSI core. Unfortunately only three out of six bound IsiA proteins are identified. Quite interestingly IsiA2 is found to be a chimeric protein with an N-terminal domain homologous to the PsbC subunit of PSII and a C-terminal domain homologous to the PsaL subunit of PSI, which facilitates PSI oligomerization. The PsaL-like domain is located in the site normally occupied by PsaL thus preventing the formation of PSI dimers and tetramers. This is a key finding of the present study, on the basis of which the authors propose that the binding of IsiA2 to the PsaL site prevents PSI oligomerization and facilitates the assembly of the PSI-IsiA complex. In a way this is a conformation of a somewhat similar mechanism earlier proposed for *Leptolyngbya* – another filamentous cyanobacterium, expressing multiple *isiA* genes under iron starvation.

Finally, the authors have probed the fluorescence kinetics in the isolated complex and identified a time constant of 55 ps, representing energy transfer from IsiA to PSI, showing that the bound IsiA complexes have (unsurprisingly) a functional role as light-harvesting antenna.

The study is conducted to the appropriate standard and the results and data analysis are reliable and well documented. This is a significant contribution across several science areas.

For the most part, the interpretations and conclusions are warranted. An exception might be the conclusion about a 707-nm band in the fluorescence decay-associated spectra that it represents excitation-energy quenching (page 16 and page 17, bottom). This statement is contentious and not entirely warranted by the data. It is possible that the fluorescence decay is due to energy transfer to lower-energy states even though the spectrum shows no negative amplitudes. Similar decay components have been observed in PSI monomers too. There is substantial literature on IsiA with regard to excitation quenching.

Further questions and comments:

- It is unfortunate that the authors do not discuss the presence (or lack) of any other classes of PSI-IsiA particles in the cryoEM images. Previous studies have shown a remarkable variety of PSI-IsiA complexes. Is this not the case here? Are there any suspected significant particles among the 50 classes of 2D particles identified in the cryoEM images?
- Another curious result is that the PsaK subunit found normally in PSI is substituted by Unknown in the present structure. This result, however, is not mentioned in the discussion or conclusions. Is it because the authors are unconvinced of the validity of their finding?
- The IsiA expression levels are given relative to iron-rich growth conditions. Is this reference reliable considering that IsiA is not normally expressed under such conditions and is it not introducing large variance in the quantification?
- Line 116: ref 31 citation seems irrelevant (it's about N deficiency not Fe).
- Line 358: ref 39 citation is irrelevant (it's about PSII).

- Line 366: Is it not possible that the 686-nm band in the 3.9 FDA spectrum reflects some IsiA complexes in the sample that are not connected to PSI as opposed to uncoupled Chls in IsiA?

- The binding positions of IsiA in Anabaena and other cyanobacteria are compared on page 18 and Supplementary Fig. 10. Perhaps it's worth to note that the IsiA locations to the outside of PsaB in the other species allow the formation of complete rings around trimeric PSI (which don't exist in Anabaena), whereas positions 1-3 in Anabaena would clash with a trimeric PSI.

The following are suggestions for improving the readability of the manuscript:

- The binding of the PsaL-like domain of IsiA to PSI is a key result in this study. Accordingly, Supplementary Fig. 5 or at least some part or version of it seems more appropriate as a main figure.

- Line 75: "isiA1, isiA2, ..." could be replaced with "isiA1-5" without loss of meaning.

- Line 112: "as detected by mass spectrometry analysis (Supplementary Fig. 1c)" – suggests that Fig. 1c shows MS results and it in fact shows SDS-PAGE.

- Line 172: What is the meaning of "significantly different" in the context of a quantitative aminoacid comparison?

- Line 175, 211: It is difficult to find the respective Chl ligands in the long lists. It would be better to write them as pairs Chl-ligand or in a table.

- Line 191: "original absence" probably means "genuine"

- Line 224: "277-326 Ca atoms" should be "Ca atoms 277-326"

- Line 345: "which have been denoted" – "which has been denoted"

- Line 374: For consistency IsiA-5 should be IsiA1-5

- Line 376: "occur" – "occurs"

- Line 395 "temporally" ?

Finally, the detailed text, manuscript organization and figures are not typical of this journal. The information density of Fig. 6 is much lower than the rest.

Reviewer #3 (Remarks to the Author):

Nagao et al. present a first structure of a PSI-IsiA supercomplex from Anabaena as determined by single particle cryoEM. To achieve this the authors overcame problems in sample/cryo-grid preparation by collecting a very large data set of more than 20,000 movies using a state-of-the-art instrument equipped with a cold field emission gun and energy filter. The structure clearly shows six IsiA complexes attached to a PSI monomer that surprisingly lacks the for PSI oligomerization important PsaL subunit. Their finding that the PsaL subunit is replaced by the c-terminal extension of IsiA2

provides a neat structural basis for PSI monomerization in the PSI-IsiA supercomplex and adds a fascinating new layer of regulation for the oligomeric state of PSI in cyanobacteria. Finally, the authors propose an assembly pathway for both the IsiA complexed PSI monomer and the IsiA free PSI tetramer. This hypothesis is both reasonable on the basis of what is currently known and, importantly, it is also testable.

I recommend the publication of this manuscript after a few minor revisions.

Minor points to consider before publication:

The structural data of the key finding that the IsiA2 c-terminal extension acts as a stand-in for the PsaL subunit is not well presented. Some readers might suspect that the quality of the map for this part of the supercomplex is insufficient for correct tracing and assignment of the IsiA2 complex. This is especially important since other PSI attached IsiA complexes are actually only present as weak, hard to model density. This problem in presentation could be easily solved by adding a main figure that clearly visualizes map/model of the relevant domain of the supercomplex. Connected to this, the authors might discuss briefly the possible reasons of why the six PSI attached IsiA complexes do exhibit such strong differences in map quality. Is the underlying reason partial occupancy or rather floppiness?

The authors state that the presence of the IsiA2 c-terminal extension is preventing oligomer formation by replacing PsaL. Since the structure of the Anabaena PSI tetramer is known, a visual that shows the structural consequence of having the IsiA2 c-terminal extension present instead of PsaL would be helpful.

As evidenced by the huge amount of image data taken and the less than ideal monodispersity of the complexes shown in Supp.Fig. 2, the authors clearly struggled to obtain cryo-grids of high quality with an even distribution of the PSI-IsiA supercomplex. In a way this is an interesting aspect of the PSI-IsiA supercomplex that should not be 'hidden' to the interested reader. Please mention and discuss the possible reasons for these experimental difficulties.

All studies on detergent solubilized membrane complexes suffer from the possible artifacts that might be caused by the detrimental effects of membrane solubilization and purification procedures. Consequently some readers of this study might suspect that the monomeric form of PSI in the PSI-IsiA is an artifact as has been suspected for monomeric forms of cyanobacterial PSI from other sources. Please discuss these issues also in the light of the AFM study on native cyanobacterial thylakoid membranes containing non-solubilized PSI-IsiA supercomplexes by Long-Sheng Zhao et al. Nat. Plants 2020.

In this study the expression of IsiA was induced by iron-deficiency, however, the introduction lacks an explanation of why this happens. Please add a brief description of the ecological background of iron-deficiency induced IsiA expression.

Is Anabaena IsiA2 the only IsiA with an extended c-terminus?

Line 99: "unclosed supercomplex" is not clear

Line 127: this might be a good place to insert a section on the strong differences in density between the different parts of the PSI-IsiA supercomplex, which strongly impacts on the modelling of the presented structure.

Line 134: unclear usage of the term "identified" which is supposedly expressing the circumstance that the density map was too weak to model an IsiA complex which can be assigned to one of the four IsiA gene products.

Line 153: instead of using "a part of" please just name it

Line 182: please state clearly, if these differences are for real, or just differences in the ability to model

Line 219: again an awkward use of "identify"

Line 304: since you do not mention lab names throughout your manuscript, please refrain from doing this here

Line 372: the word "interactions" suggests physical contact, however, you likely rather want to say something like "distance and geometry that allows for efficient EET"
-> connected to this, please check Supp.Fig. 9 for the distance labels; to my knowledge distances for excitation energy transfer between chromophores are measured center-to-center and not edge-to-edge

Fig. 6: are the FDA spectra shown from a single measurement or is it one measurement representative of n measurements?

Supp. Fig 12: map is hardly visible, I suggest to use a different, more clear visual and why not use the whole A4 page, i.e. top-down arrangement of the three IsiA?

Reviewer #4 (Remarks to the Author):

1. The major concern is the identification of IsiA2. In Supplementary Figs. 5 & 6, the authors showed that the densities match well with several characteristic residues of IsiA1 and IsiA2 to support their assignments. However, assignments based solely on EM density are not conclusive. Based on Supplementary Fig. 1A, the transcription of IsiA2 is much lower than that of other IsiAs under iron-deficient conditions. And as shown in Supplementary Fig. 1C, the IsiA2 band is very faint, indicating that the protein content of IsiA2 in the complex is extremely low compared to those of IsiA1, IsiA3/5, and PsaD/PsaF. Moreover, according to Supplementary Fig. 3C, the map in IsiA part shows a resolution of about 3.5 angstrom or even lower. The low-resolution of IsiAs is also indicated by their low Q-score shown in Supplementary Table 2. Therefore, while the possibility that IsiA2 is indeed present in this complex cannot be completely ruled out, it is more likely that the currently assigned IsiA2 is actually a canonical IsiA plus a PsaL. Could the authors comment on this?

2. Another unusual feature of this structure is that the two IsiA1 subunits (IsiA1-4 and IsiA1-5), despite being the same gene product, have drastically different pigment compositions. The authors provided a potential explanation (the weaker densities of IsiA1-4 compared with that of IsiA1-5), which is not convincing. There is no much difference between the resolutions of the two subunits in Supplementary Fig. 3C.

3. PDB validation report: "the value from deposited half-maps intersecting FSC 0.143 CUT-OFF 3.11 differs from the reported value 2.62 by more than 10 %". This issue should be fixed. In addition, the fitness between model and EM map is rather poor for all IsiAs, with the majority of residues appear poor fit. The authors should check whether the final cryo-EM map was post-processed with a proper mask and b-factor.

4. Page 4, second Paragraph: LHC usually refers to peripheral antenna proteins containing three transmembrane helices, thus it's better to replace water-soluble LHCs with water-soluble antennae or something similar.

Reviewer #1 (Remarks to the Author):

The manuscript by Nagao et al. describes high-resolution structures of photosystem I from the cyanobacterium *Anabaena* grown under iron-deficient condition. The structure of PSI-IsiA supercomplex was solved by cryo-electron microscopy (cryo-EM) single-particle analysis. The results showed that PSI-IsiA supercomplex consists of six IsiA subunits encoded by different *isiA* genes associated at one site of a monomeric PSI core, forming an unclosed, monomeric PSI-IsiA supercomplex. This is a novel finding showing once again the plasticity of PSI in various organisms responding to environmental stress.

The manuscript is very interesting and accomplished by highly capable scientists. For the benefit of the work and the readers it could be shortened. I would recommend accepting it for publication after addressing some obstacles.

First of all, we thank you very much for your highly positive comments on our manuscript. Based on your comments and suggestions, we have modified our manuscript, which are listed below.

Comment 1:

The FSC curves of PSI-IsiA presented in Supplementary Fig. 3a. is not of a high quality we used to see from the corresponding author laboratory. This indicates some local structural problems. Indeed, throughout the manuscript it is stated that interactions could not be concluded because of poor local density.

Author reply 1:

We agree that the FSC curves presented in Supplementary Fig. 3a is not of a high quality, and this may be caused by poor local resolutions especially in the peripheral IsiA subunits. We have stated this point in the manuscript (lines 192-193) that “a solid conclusion regarding the detailed interactions within this supercomplex has to wait until a higher resolution structure is obtained”.

Comment 2:

The interactions of chlorophylls with ligands are discussed in detail and I would recommend shorten those descriptions that have no bearing on the mechanism. The relevant PDBs will be published and the readers that are interested in specific aspects will be able to obtain the information.

Author reply 2:

Based on your comments, we dramatically modified the section regarding the interactions between chlorophylls with ligands. Instead, we added new sections of “Structure of IsiA2-1”, “Structures of

IsiA1-4 and IsiA1-5”, and “Structures of IsiA-2, IsiA-3, and IsiA-6” to the Result section in the revised manuscript, to describe the structures and their different differences among the six IsiAs more clearly. Also, new Fig. 2 and 3 were added. Details about correlation between Chls and ligands are shown in Supplementary Fig. 6, 8 and listed in Supplementary Table 4.

Comment 3:

A previous study from the same group using two-dimensional BN/SDS-PAGE analysis of thylakoids from *Anabaena* grown under the iron-deficient condition detected only IsiA1 subunit in the fraction of the PSI-IsiA supercomplex (Nagao, R. et al. 2021). This is a prime example for the fact that blue-native gels produce more artifacts than facts. The explanation for this discrepancy is not convincing. Accordingly, remove panel 1 d from supplementary Figure 1. and minimize the conclusion drawn throughout the manuscript relying on this technique.

Author reply 3:

According to your comments, we removed panel 1d from supplementary Fig. 1, and minimized the description about BN-PAGE, especially the first paragraph of “Expressions of the *Anabaena* IsiA family and their structures” in the Discussion section (lines 252-259).

Comment 4:

The discovery of C-terminal PsaL-like domain in IsiA is highly interesting and merits publication of this work.

Author reply 4:

Thank you. To focus on the topics of IsiA2-1, we added a new section “Structure of IsiA2-1” to the Result section in the revised manuscript to describe the structure of IsiA2-1 more clearly. In addition, a new Fig. 2 was added to depict a connection of the IsiA N-terminal domain with the C-terminal PsaL-like domain in IsiA2-1 (Fig. 2e).

Comment 5:

Minor corrections

82 related to isiA1 of *Anabaena*30. Immunological analysis of IsiAs detected one band in Immunological analysis of IsiAs detected identified one band in thylakoids of *Anabaena* prepared from the iron-deficient cells.

Author reply 5:

To avoid misleading, we removed this sentence from the revised manuscript. This is related to your

comment 3 as described above.

Comment 6:

Please clarify

84 In addition, two- dimensional blue-native (BN)/SDS-PAGE analysis together with mass spectrometry detected the IsiA1 subunit in a PSI fraction located near the band of the PSII dimer, showing the formation of a PSI-IsiA1 supercomplex in Anabaena30

Author reply 6:

To avoid misleading, we removed this sentence from the revised manuscript. This is related to your comment 3 as described above.

Comment 7:

Correct

152 the substitution of PsaK with Unknown or inhibit the expression of PsaK.

Author reply 7:

We removed the sentence from the revised manuscript.

Reviewer #2 (Remarks to the Author):

This submission presents surprising and interesting discoveries about the structural interactions in a complex of photosystem I with iron-stress-induced protein A (PSI-IsiA) in the filamentous nitrogen-fixing cyanobacterium *Anabaena* sp. PCC 7120. As Fe is an essential nutrient, photosynthetic organisms and cyanobacteria in particular have evolved various mechanisms for Fe homeostasis and acclimation to Fe-limiting conditions. One mechanism among cyanobacteria is the synthesis of IsiA proteins that are homologous to the PSII core antenna protein PsbC. IsiA are known to form various ring-like structures around PSI, and to enhance the PSI absorption cross section, which compensates for the reduced synthesis of phycobiliproteins and PSI under Fe limitation. This contribution reveals the first medium-resolution structure of PSI-IsiA from *Anabaena* and one of a very few structures available overall – the others being from *Thermotichus vulcanus* published in 2020 by the same group and from *Synechocystis* sp. (Toporik et al. 2019). Both of these structures contain identical copies of IsiA forming a ring around a PSI trimer, whereas the *Anabaena* complex is rather special, containing different IsiA subunits occupying different sites around a monomeric PSI complex. These studies bring into light an important adaptation mechanism and design principles by which cyanobacteria regulate the antenna size of the photosynthetic reaction centres.

The present structure is probably not the definitive one, as not all subunits and cofactors can be identified due to insufficient resolution. Nonetheless, the authors reveal some novel aspects deserving attention. They show that multiple *isiA* genes are expressed in *Anabaena* under iron-limiting conditions and the respective IsiA proteins occupy specific binding sites at the periphery of the PSI core. Unfortunately only three out of six bound IsiA proteins are identified. Quite interestingly IsiA2 is found to be a chimeric protein with an N-terminal domain homologous to the PsbC subunit of PSII and a C-terminal domain homologous to the PsaL subunit of PSI, which facilitates PSI oligomerization. The PsaL-like domain is located in the site normally occupied by PsaL thus preventing the formation of PSI dimers and tetramers. This is a key finding of the present study, on the basis of which the authors propose that the binding of IsiA2 to the PsaL site prevents PSI oligomerization and facilitates the assembly of the PSI-IsiA complex. In a way this is a conformation of a somewhat similar mechanism earlier proposed for *Leptolyngbya* – another filamentous cyanobacterium, expressing multiple *isiA* genes under iron starvation.

Finally, the authors have probed the fluorescence kinetics in the isolated complex and identified a time constant of 55 ps, representing energy transfer from IsiA to PSI, showing that the bound IsiA complexes have (unsurprisingly) a functional role as light-harvesting antenna.

The study is conducted to the appropriate standard and the results and data analysis are reliable and well documented. This is a significant contribution across several science areas.

First of all, we thank you very much for your detailed and highly positive evaluation of our manuscript. Based on your comments and suggestions, we have modified our manuscript which are listed below.

Comment 1:

For the most part, the interpretations and conclusions are warranted. An exception might be the conclusion about a 707-nm band in the fluorescence decay-associated spectra that it represents excitation-energy quenching (page 16 and page 17, bottom). This statement is contentious and not entirely warranted by the data. It is possible that the fluorescence decay is due to energy transfer to lower-energy states even though the spectrum shows no negative amplitudes. Similar decay components have been observed in PSI monomers too. There is substantial literature on IsiA with regard to excitation quenching.

Author reply 1:

If negative magnitudes originating from low-energy Chls are much low, it is possible that the fluorescence decay is due to energy transfer to lower-energy states even though the spectrum shows no negative amplitudes. In Figure 7, positive amplitudes in the third and fourth FDA spectra are ~1 and ~0.2, respectively, and negative magnitudes corresponding to these positive amplitudes were observed only in the first FDA spectrum. Therefore, the energy transfer to lower-energy states is expressed by the first FDA spectrum. Even if the energy transfer to the lower-energy states occurs in the second FDA spectrum, its contribution should be negligibly small compared with that in the first FDA spectrum. Moreover, the fluorescence decays at 690 and 707 nm in the 120-ps FDA spectrum did not exist in a PSI monomer (Nagao et al., JPCB, 2020, 124, 1949–1954), indicating that these two decays originate from IsiAs. Therefore, we propose that excitation-energy quenching may occur at Chls fluorescing at 690 and 707 nm.

Comment 2:

Further questions and comments:

- It is unfortunate that the authors do not discuss the presence (or lack) of any other classes of PSI-IsiA particles in the cryoEM images. Previous studies have shown a remarkable variety of PSI-IsiA complexes. Is this not the case here? Are there any suspected significant particles among the 50 classes of 2D particles identified in the cryoEM images?

Author reply 2:

As far as the particle images are concerned, it seemed that no other types of PSI-IsiA are observed (see the figure of Class 2D below). However, it is possible that other types of PSI-IsiAs are also present in *Anabaena*, and these types are separated from the sample used in the present study or are not picked up in the present study.

Comment 3:

- Another curious result is that the PsaK subunit found normally in PSI is substituted by Unknown in the present structure. This result, however, is not mentioned in the discussion or conclusions. Is it because the authors are unconvinced of the validity of their finding?

Author reply 3:

We are convinced of the validity of our finding, so have added a paragraph in the Discussion section to describe the difference of PsaK between *Anabaena* and other cyanobacteria and its possible consequences (lines 406-413).

Comment 4:

- The IsiA expression levels are given relative to iron-rich growth conditions. Is this reference reliable considering that IsiA is not normally expressed under such conditions and is it not introducing large variance in the quantification?

Author reply 4:

As noted by the reviewer, the expression of *isiA* is low under iron-rich growth conditions. However, the quantity of qRT-PCR is sufficient to quantify such low levels of gene expression. Because *Anabaena* has siderophores that can uptake iron even at very low concentrations, residual iron in the

medium can support the growth of *Anabaena*. What is important to us is to compare the different expression levels of the different *isiA* genes under the iron-deficient condition, even a large variation exists in the quantification of expression level under iron-rich condition, this will not affect our conclusions. The *isiA* expression was greatly induced in all experiments, indicating that sufficient amounts of IsiA proteins were produced to form the PSI-IsiA supercomplex.

Comment 5

- Line 116: ref 31 citation seems irrelevant (it's about N deficiency not Fe).

Author reply 5:

We revised it, thank you.

Comment 6:

- Line 358: ref 39 citation is irrelevant (it's about PSII).

Author reply 6:

We revised it, thank you.

Comment 7:

- Line 366: Is it not possible that the 686-nm band in the 3.9 FDA spectrum reflects some IsiA complexes in the sample that are not connected to PSI as opposed to uncoupled Chls in IsiA?

Author reply 7:

We agree with your comments. The possibility you mentioned remains; therefore, we added the sentence “and/or weak interactions between IsiAs and PSI exist in the supercomplex” to the revised manuscript (lines 353-354).

Comment 8:

- The binding positions of IsiA in *Anabaena* and other cyanobacteria are compared on page 18 and Supplementary Fig. 10. Perhaps it's worth to note that the IsiA locations to the outside of PsaB in the other species allow the formation of complete rings around trimeric PSI (which don't exist in *Anabaena*), whereas positions 1-3 in *Anabaena* would clash with a trimeric PSI.

Author reply 8:

Thank you, we described these contents in the section “Structural comparisons of IsiAs between *Anabaena* and other cyanobacteria” of the Discussion section (pages 17-18).

Comment 9:

The following are suggestions for improving the readability of the manuscript:

- The binding of the PsaL-like domain of IsiA to PSI is a key result in this study. Accordingly, Supplementary Fig. 5 or at least some part or version of it seems more appropriate as a main figure.

Author reply 9:

We agreed with your comment, and added the content of “Structure of IsiA2-1” as a main Figure 2 to the revised manuscript. To focus on the topics of IsiA2-1, we added a new section “Structure of IsiA2-1” to the Result section in the revised manuscript.

Comment 10:

- Line 75: “isiA1, isiA2, ...” could be replaced with “isiA1-5” without loss of meaning.

Author reply 10:

We modified it.

Comment 11:

- Line 112: “as detected by mass spectrometry analysis (Supplementary Fig. 1c)” – suggests that Fig. 1c shows MS results and it in fact shows SDS-PAGE.

Author reply 11:

We removed the words “as detected by mass spectrometry analysis” from the revised manuscript.

Comment 12:

- Line 172: What is the meaning of “significantly different” in the context of a quantitative amino acid comparison?

Author reply 12:

This sentence does not represent a quantitative comparison. Simply, we show difference in amino acid compositions between F45/W47/K279/G281/V282/T283 in IsiA1 and the corresponding amino acids in the other IsiAs, and removed the word “significantly”. For example, IsiA1-F45 vs. IsiA2-M45, IsiA3-T46, IsiA5-T49. To explain this, we inserted the words “, e.g., IsiA1-F45 vs. IsiA2-M45, IsiA3-T46, IsiA5-T49, etc.” into the corresponding section (line 178).

Comment 13:

- Line 175, 211: It is difficult to find the respective Chl ligands in the long lists. It would be better to write them as pairs Chl-ligand or in a table.

Author reply 13:

We agree with your comment. Details about correlation between Chls and ligands are shown in Supplementary Fig. 6, 8 and listed in Supplementary Table 4.

Comment 14:

- Line 191: “original absence” probably means “genuine”

Author reply 14:

We changed the sentence as “Alternatively, some Chls may be naturally absent in IsiA1-4.” (line 190).

Comment 15:

- Line 224: “277-326 Ca atoms” should be “Ca atoms 277-326”

Author reply 15:

According to a lot of papers, we would like to keep this format “277-326 Ca atoms”, and modified the original words to “a total of 277-326 Ca atoms”.

Comment 16:

- Line 345: “which have been denoted” – “which has been denoted”

Author reply 16:

We modified it.

Comment 17:

- Line 374: For consistency IsiA-5 should be IsiA1-5

Author reply 17:

Yes, thank you; we modified it.

Comment 18:

- Line 376: “occur” – “occurs”

Author reply 18:

We modified it.

Comment 19:

- Line 395 “temporally” ?

Author reply 19:

We removed it.

Comment 20:

Finally, the detailed text, manuscript organization and figures are not typical of this journal. The information density of Fig. 6 is much lower than the rest.

Author reply 20:

We dramatically improved our manuscript. Figure 6 depicts several curves which contain much information regarding the time-resolved fluorescence emission, so we kept it in its original form.

Reviewer #3 (Remarks to the Author):

Nagao et al. present a first structure of a PSI-IsiA supercomplex from *Anabaena* as determined by single particle cryoEM. To achieve this the authors overcame problems in sample/cryo-grid preparation by collecting a very large data set of more than 20,000 movies using a state-of-the-art instrument equipped with a cold field emission gun and energy filter. The structure clearly shows six IsiA complexes attached to a PSI monomer that surprisingly lacks the for PSI oligomerization important PsaL subunit. Their finding that the PsaL subunit is replaced by the c-terminal extension of IsiA2 provides a neat structural basis for PSI monomerization in the PSI-IsiA supercomplex and adds a fascinating new layer of regulation for the oligomeric state of PSI in cyanobacteria. Finally, the authors propose an assembly pathway for both the IsiA complexed PSI monomer and the IsiA free PSI tetramer. This hypothesis is both reasonable on the basis of what is currently known and, importantly, it is also testable.

I recommend the publication of this manuscript after a few minor revisions.

First of all, we thank you very much for your highly positive and detailed evaluation of our manuscript. Based on your comments and suggestions, we have modified our manuscript which are listed below.

Minor points to consider before publication:

Comment 1:

The structural data of the key finding that the IsiA2 c-terminal extension acts as a stand-in for the PsaL subunit is not well presented. Some readers might suspect that the quality of the map for this part of the supercomplex is insufficient for correct tracing and assignment of the IsiA2 complex. This is especially important since other PSI attached IsiA complexes are actually only present as weak, hard to model density. This problem in presentation could be easily solved by adding a main figure that clearly visualizes map/model of the relevant domain of the supercomplex. Connected to this, the authors might discuss briefly the possible reasons of why the six PSI attached IsiA complexes do exhibit such strong differences in map quality. Is the underlying reason partial occupancy or rather floppiness?

Author reply 1:

We agree with your comment about the structure of IsiA2-1. We added new sections of “Structure of IsiA2-1”, “Structures of IsiA1-4 and IsiA1-5”, and “Structures of IsiA-2, IsiA-3, and IsiA-6” to the Result section in the revised manuscript. New Fig. 2 and 3 were also added. As you may see from the

newly added Fig. 2, the map quality for IsiA2-1 is good enough to allow the tracing of the sequence. Also, the N-terminal and C-terminal domains of the polypeptide is well connected, resulting in a single polypeptide. Thus, we are confident about the assignment of IsiA2-1. In addition, we also dramatically modified the section of chlorophylls and ligands. Details about correlation between Chls and ligands are shown in Supplementary Fig. 6, 8 and listed in Supplementary Table 4.

About your second question, the different map quality observed in the peripheral region of PSI-IsiA often occurs in photosystem-LHC supercomplexes. For example, the structures of LHCII in PSII-LHCII from land plants, green algae, and diatoms all have a low map quality. This may be due to a high structural flexibility of the LHC region compared with photosystem cores or partial dissociation of LHCs from supercomplexes. To explain this, we added a sentence “This may be due to a high structural flexibility of the IsiA region compared with PSI and/or partial dissociation of IsiAs from PSI-IsiA, leading to a lower occupancy and hence a lower resolution.” to the Result section in the revised manuscript (lines 129-131).

Comment 2:

The authors state that the presence of the IsiA2 c-terminal extension is preventing oligomer formation by replacing PsaL. Since the structure of the Anabaena PSI tetramer is known, a visual that shows the structural consequence of having the IsiA2 c-terminal extension present instead of PsaL would be helpful.

Author reply 2:

As summarized in Fig. 8, oligomeric formation of PSI tetramer may be suppressed by binding of IsiA2 to PSI instead of PsaL. This seems to be caused by the N-terminal IsiA domain of IsiA2. To avoid misleading, we modified the corresponding sentence in the revised manuscript (line 300).

If only the C-terminal PsaL-like domain of IsiA2 is expressed by genetic manipulation, a tetrameric formation of PSI might occur under iron-deficient conditions. Such study will be performed in the future.

Comment 3:

As evidenced by the huge amount of image data taken and the less than ideal monodispersity of the complexes shown in Supp. Fig. 2, the authors clearly struggled to obtain cryo-grids of high quality with an even distribution of the PSI-IsiA supercomplex. In a way this is an interesting aspect of the PSI-IsiA supercomplex that should not be 'hidden' to the interested reader. Please mention and discuss the possible reasons for these experimental difficulties.

Author reply 3:

Photosystem-LHC supercomplexes are easily dissociated during preparations of the protein complexes and cryo-EM grids. So, a large amount of image data is often required for 3D reconstruction of supercomplexes. To explain this, we added the sentences “It is known that photosystem-LHC supercomplexes are easily dissociated during preparations of the samples and/or cryo-EM grids. Therefore, acquiring a large amount of image data is often required for 3D reconstruction of supercomplex structures.” to the method section of “Cryo-EM data collection.” in the revised manuscript (lines 536-539).

Comment 4:

All studies on detergent solubilized membrane complexes suffer from the possible artifacts that might be caused by the detrimental effects of membrane solubilization and purification procedures. Consequently some readers of this study might suspect that the monomeric form of PSI in the PSI-IsiA is an artifact as has been suspected for monomeric forms of cyanobacterial PSI from other sources. Please discuss these issues also in the light of the AFM study on native cyanobacterial thylakoid membranes containing non-solubilized PSI-IsiA supercomplexes by Long-Sheng Zhao et al. *Nat. Plants* 2020.

Author reply 4:

According to your comment, we added a new paragraph to the Discussion section of “Molecular assembly of the PSI-monomer-IsiA supercomplex” to the revised manuscript (lines 309-317), to describe that the PSI-monomer-IsiA structure obtained in the present study may represent one of the PSI-IsiA forms *in vivo*.

Comment 5:

In this study the expression of IsiA was induced by iron-deficiency, however, the introduction lacks an explanation of why this happens. Please add a brief description of the ecological background of iron-deficiency induced IsiA expression.

Author reply 5:

We agree with your comment, and modified the second paragraph of Introduction section in the revised manuscript (lines 64-68) to describe the expression of IsiA under iron-deficient conditions.

Comment 6:

Is Anabaena IsiA2 the only IsiA with an extended c-terminus?

Author reply 6:

Yes, *Anabaena* has four *isiA* genes, among which IsiA2 has the only IsiA with a C-terminal extension. In other cyanobacteria, for example, *Leptolyngbya* sp. strain JSC-1 has five *isiA* genes, among which IsiA4 has an extra C-terminal PsaL-like domain. These contents were described in the text (lines 74-90).

Comment 7:

Line 99: "unclosed supercomplex" is not clear

Author reply 7:

We removed it.

Comment 8:

Line 127: this might be a good place to insert a section on the strong differences in density between the different parts of the PSI-IsiA supercomplex, which strongly impacts on the modelling of the presented structure.

Author reply 8:

According to your comment, we added the sentence “, although the peripheral region of IsiAs has a relatively low resolution (Supplementary Fig. 3c)” to the section “Overall structure of the PSI-IsiA supercomplex” of Results in the revised manuscript (lines 118-119).

Comment 9:

Line 134: unclear usage of the term "identified" which is supposedly expressing the circumstance that the density map was too weak to model an IsiA complex which can be assigned to one of the four IsiA gene products.

Author reply 9:

We changed the word “identified” to “assigned” in the revised manuscript.

Comment 10:

Line 153: instead of using "a part of" please just name it

Author reply 10:

The words “a part of” were changed to “C-terminal domain” in the revised manuscript.

Comment 11:

Line 182: please state clearly, if these differences are for real, or just differences in the ability to model

Author reply 11:

According to your comment, we largely modified the Result section of “Structures of IsiA1-4 and IsiA1-5” in the revised manuscript (page 9), to describe the differences in the structure and cofactors between IsiA1-4 and IsiA1-5. The differences in the Chl numbers between IsiA1-4 and IsiA1-5 could be real or the ability to model, or both, and we stated this in the revised text.

Comment 12:

Line 219: again an awkward use of "identify"

Author reply 12:

We modified it as “remaining three” in the revised manuscript.

Comment 13:

Line 304: since you do not mention lab names throughout your manuscript, please refrain from doing this here

Author reply 13:

We modified it as “Bryant and co-workers” in the revised manuscript.

Comment 14:

Line 372: the word "interactions" suggests physical contact, however, you likely rather want to say something like "distance and geometry that allows for efficient EET"

-> connected to this, please check Supp. Fig. 9 for the distance labels; to my knowledge distances for excitation energy transfer between chromophores are measured center-to-center and not edge-to-edge

Author reply 14:

If excitation-energy transfer is based on Förster theory, an important assumption is that a chromophore-chromophore distance including edge-to-edge is sufficiently larger than a size of chromophore molecule (in case of Chl, $\sim 8.8 \text{ \AA}$) for the energy transfer to occur. This assumption cannot be necessarily adapted to all chromophore-chromophore pairs; therefore, it is very difficult to propose perfect excitation-energy transfer using either center-to-center or edge-to-edge based on structural data. In contrast, as structural biologists, we focus on physical contacts among atoms in the neighboring chromophores. For these reasons, we showed edge-to-edge distances among pigments as the shortest distances between chromophores in both the present study and our previous studies.

Comment 15:

Fig. 6: are the FDA spectra shown from a single measurement or is it one measurement representative of n measurements?

Author reply 15:

We measured 3 sets of TRF spectra, and FDA spectra were analyzed by using DATA with the longest accumulation time. We added this into the revised manuscript (line 508).

Comment 16:

Supp. Fig 12: map is hardly visible, I suggest to use a different, more clear visual and why not use the whole A4 page, i.e. top-down arrangement of the three IsiA?

Author reply 16:

According to your comment, we modified supplementary Fig. 12 in the revised manuscript to show the maps more clearly.

Reviewer #4 (Remarks to the Author):

Comment 1:

1. The major concern is the identification of IsiA2. In Supplementary Figs. 5 & 6, the authors showed that the densities match well with several characteristic residues of IsiA1 and IsiA2 to support their assignments. However, assignments based solely on EM density are not conclusive. Based on Supplementary Fig. 1A, the transcription of IsiA2 is much lower than that of other IsiAs under iron-deficient conditions. And as shown in Supplementary Fig. 1C, the IsiA2 band is very faint, indicating that the protein content of IsiA2 in the complex is extremely low compared to those of IsiA1, IsiA3/5, and PsaD/PsaF. Moreover, according to Supplementary Fig. 3C, the map in IsiA part shows a resolution of about 3.5 angstrom or even lower. The low-resolution of IsiAs is also indicated by their low Q-score shown in Supplementary Table 2. Therefore, while the possibility that IsiA2 is indeed present in this complex cannot be completely ruled out, it is more likely that the currently assigned IsiA2 is actually a canonical IsiA plus a PsaL. Could the authors comment on this?

Author reply 1:

First of all, we thank you very much for your evaluation of our manuscript. To focus on the topics of IsiA2-1, we added a new section “Structure of IsiA2-1” to the Result section in the revised manuscript. In addition, a new Fig. 2e was built to represent a connection of the N-terminal IsiA domain with the C-terminal PsaL-like domain in IsiA2-1. Only one IsiA2 subunit exists in the PSI-IsiA structure. It is possible that other *isiA* genes (*isiA1*, *isiA3* and *isiA5*) are present in multiple copies in the PSI-monomer-IsiA structure solved in the present study as well as in some other forms of PSI-IsiAs whose structure has not been solved; therefore, it seems to be reasonable that the mRNA and protein expression of IsiA2 is lower than those of the other IsiAs. The difference in the characteristic amino acids between PsaL and the C-terminal PsaL-like domain of IsiA2 is shown in a new Fig. 2c,d. These observations show that PsaL is lacking in the present PSI-IsiA structure and the C-terminal PsaL-like domain of IsiA2 is positioned at the site of PsaL. As shown in the new Fig. 2e, the N-terminal and C-terminal domains of the polypeptide are well connected, indicating that it is a single polypeptide rather than a canonical IsiA plus a PsaL. To explain these contents, we dramatically improved the revised manuscript.

Comment 2:

2. Another unusual feature of this structure is that the two IsiA1 subunits (IsiA1-4 and IsiA1-5), despite being the same gene product, have drastically different pigment compositions. The authors provided a potential explanation (the weaker densities of IsiA1-4 compared with that of IsiA1-5), which is not convincing. There is no much difference between the resolutions of the two subunits in Supplementary

Fig. 3C.

Author reply 2:

We agree with your comments. For model building, most structural biologists use a Coot software with a cryo-EM map of Post-process but not local resolution. The Coot software shows different map quality between IsiA1-4 and IsiA1-5, although their local resolution maps seems to have a similar quality. In addition, the different map quality observed in the peripheral region of PSI-IsiA often occurs in photosystem-LHC supercomplexes. For example, the structures of LHCII in PSII-LHCII from land plants, green algae, and diatoms all have a low map quality. This may be due to a high structural flexibility of LHC region compared with photosystem cores or partial dissociation of LHCs from supercomplexes. To explain this, we added sentences “This may be due to a high structural flexibility of the IsiA region compared with PSI and/or partial dissociation of IsiAs from PSI-IsiA, leading to a lower occupancy and hence a lower resolution.” to the Result section in the revised manuscript (lines 129-131).

Comment 3:

3. PDB validation report: “the value from deposited half-maps intersecting FSC 0.143 CUT-OFF 3.11 differs from the reported value 2.62 by more than 10 %”. This issue should be fixed. In addition, the fitness between model and EM map is rather poor for all IsiAs, with the majority of residues appear poor fit. The authors should check whether the final cryo-EM map was post-processed with a proper mask and b-factor.

Author reply 3:

The FSC value of our validation report represents an unmasked FSC, which is different from a masked FSC (2.62 Å). In addition, our mask and B-factor are suitable for the final reconstruction. This is because of a relevant Guinier plots and mask structures as shown in the figures below.

Comment 4:

4. Page 4, second Paragraph: LHC usually refers to peripheral antenna proteins containing three transmembrane helices, thus it's better to replace water-soluble LHCs with water-soluble antennae or something similar.

Author reply 4:

According to your comment, we removed the corresponding sentence from the revised manuscript.

REVIEWERS' COMMENTS

Reviewer #1 (Remarks to the Author):

I have no concerns about the new revised version.

Reviewer #2 (Remarks to the Author):

The revised manuscript is improved significantly with more informative illustrations and a refined and better structured text. The authors have addressed all minor and the majority of major criticisms. However, I find the rebuttal to at least one comment unsatisfactory. Mainly, the claim about excitation quenching in IsiA (comment 1):

1) Your reply that energy transfer to low-energy Chls is observed in the first FDA is correct but misses the point: The second FDA (120 ps decay of emission at 707 nm) does not necessarily represent excitation deactivation but can also be due to relaxation to a lower-energy excited state.

2) If the 55 ps FDA is energy transfer to PSI - with clear negative feature at 707 nm, then it is likely that the 707 nm emission originates from PSI.

3) Even if they are not resolved in all time-resolved fluorescence data, states emitting around 707 nm have been found in nearly every PSI. They invariably decay faster than the lowest-energy traps at low temperature.

4) Among the rich literature on IsiA, have states emitting at 707 nm been found? It is a stretch to assume, without evidence, that such drastic red shift will occur due to the interaction with PSI.

5) The putative carotenoid-chlorophyll coupling between IsiA and PSI (lines 370-375) as a quenching mechanism is merely handwaving. The paper will only be better without it.

6) It is pretty clear that the 3.9 ns decay is from IsiAs that are not energetically connected to PSI. This may be a sample issue or the case in vivo but that "weak interactions between IsiAs and PSI exist in the supercomplex" is probably not the most accurate description.

The above remarks are of minor importance and are almost irrelevant to the main conclusions of the overall interesting and significant study. Therefore, I support publication.

Reviewer #4 (Remarks to the Author):

After checking the cryo-EM map, this reviewer thinks that the authors' identification of IsiA2-1 is correct. This manuscript reports an interesting and unique PSI-IsiA structure. The revised manuscript has been greatly improved, therefore this reviewer would recommend acceptance of this manuscript after some minor revisions.

1. The IsiA-2, -3 and -6 have poorer densities, and the resolutions of these regions differ significantly from the claimed overall resolution of 2.62 angstrom. This point should be clearly stated in the manuscript.

2. Figs. S6 & S8, these two figures are not very informative. Rather than simply displaying the chlorophylls and their ligands for each IsiA, it's better to show the superposition results of IsiAs solved in the current work with IsiAs from other structures (species) reported earlier. This comparison result will provide readers a full picture of the similarities and differences of different IsiAs in terms of the protein folding and bound pigments.

3. Fig. S7b & c, please indicate which IsiA1 is shown here, or are these two regions from two different IsiA1s?

Reviewer #5 (Remarks to the Author):

Now I have read the revised manuscript of Nagao et al. and do find it greatly improved and all queries addressed.

I have no further comments and hope it will be published soon.

Merry Christmas and a Happy New Year to everyone.

Reviewer #1 (Remarks to the Author):

I have no concerns about the new revised version.

Author reply:

Thank you.

Reviewer #2 (Remarks to the Author):

The revised manuscript is improved significantly with more informative illustrations and a refined and better structured text. The authors have addressed all minor and the majority of major criticisms. However, I find the rebuttal to at least one comment unsatisfactory. Mainly, the claim about excitation quenching in IsiA (comment 1):

First of all, we thank you very much for your highly positive evaluation and comments on our manuscript. Based on your comments and suggestions, we have modified our manuscript, which are listed below.

Comment 1:

1) Your reply that energy transfer to low-energy Chls is observed in the first FDA is correct but misses the point: The second FDA (120 ps decay of emission at 707 nm) does not necessarily represent excitation deactivation but can also be due to relaxation to a lower-energy excited state.

Author reply 1:

The second FDA spectrum did not show any negative bands, indicating that if the energy transfer occurs to the low-energy forms, its contribution is smaller than the quenching, because the third and fourth FDA spectra exhibited positive bands with significant magnitudes. However, not to avoid the possibility of the relaxation to lower-energy states, we added “mainly” in line 344.

Comment 2:

2) If the 55 ps FDA is energy transfer to PSI - with clear negative feature at 707 nm, then it is likely that the 707 nm emission originates from PSI.

Author reply 2:

As stated in the text (lines 340-341), a distinct positive band at 707 nm was not observed in the *Anabaena* PSI monomer without IsiAs. Thus, we think that this band arises from the interactions between IsiAs and PSI. To prevent misreading, we replaced “at interfaces” in lines 342 with “by interactions”. Also see our answer to your comment 4.

Comment 3:

3) Even if they are not resolved in all time-resolved fluorescence data, states emitting around 707 nm have been found in nearly every PSI. They invariably decay faster than the lowest-energy traps at low temperature.

Author reply 3:

In lines 340-342, we compared the *Anabaena* PSI and *Anabaena* PSI-IsiA. To make our explanation clear, we added “*Anabaena*” and “in *Anabaena*” in lines 341-342.

Comment 4:

4) Among the rich literature on IsiA, have states emitting at 707 nm been found? It is a stretch to assume, without evidence, that such drastic red shift will occur due to the interaction with PSI.

Author reply 4:

The 707 nm band has not been found in IsiAs. Our results showed that the *Anabaena* PSI-IsiA emits the clear 707-nm band but the *Anabaena* PSI monomer does not. Therefore, we described “this clear 707-nm band occurs by interactions between IsiAs and PSI in *Anabaena*.” in lines 342-343.

Comment 5:

5) The putative carotenoid-chlorophyll coupling between IsiA and PSI (lines 370-375) as a quenching mechanism is merely handwaving. The paper will only be better without it.

Author reply 5:

The possibility that Chl-Car couplings induce energy quenching cannot be excluded in photosynthetic pigment-protein complexes. Therefore, we did not remove the corresponding sentence.

Comment 6:

6) It is pretty clear that the 3.9 ns decay is from IsiAs that are not energetically connected to PSI. This may be a sample issue or the case in vivo but that “weak interactions between IsiAs and PSI exist in the supercomplex” is probably not the most accurate description.

Author reply 6:

To prevent misreading, we removed the sentence “weak interactions between IsiAs and PSI exist in the supercomplex” from the manuscript.

Comment 7:

The above remarks are of minor importance and are almost irrelevant to the main conclusions of the overall interesting and significant study. Therefore, I support publication.

Author reply 7:

We revised our manuscript according to your comments.

Reviewer #4 (Remarks to the Author):

After checking the cryo-EM map, this reviewer thinks that the authors' identification of IsiA2-1 is correct. This manuscript reports an interesting and unique PSI-IsiA structure. The revised manuscript has been greatly improved, therefore this reviewer would recommend acceptance of this manuscript after some minor revisions.

First of all, we thank you very much for your highly positive evaluation and comments on our manuscript. Based on your comments and suggestions, we have modified our manuscript, which are listed below.

Comment 1:

1. The IsiA-2, -3 and -6 have poorer densities, and the resolutions of these regions differ significantly from the claimed overall resolution of 2.62 angstrom. This point should be clearly stated in the manuscript.

Author reply 1:

Based on your comments, we added "as the cryo-EM map in these regions has a lower resolution than that of the overall resolution;" into the revised manuscript (lines 130-131).

Comment 2:

2. Figs. S6 & S8, these two figures are not very informative. Rather than simply displaying the chlorophylls and their ligands for each IsiA, it's better to show the superposition results of IsiAs solved in the current work with IsiAs from other structures (species) reported earlier. This comparison result will provide readers a full picture of the similarities and differences of different IsiAs in terms of the protein folding and bound pigments.

Author reply 2:

Our previous paper showed that the amino acid ligands of Chls are conserved among IsiAs observed in the PSI-IsiA structures of three types of cyanobacteria (Figure S5 in Nagao et al., BBA 2021). We also proposed that in *Anabaena*, putative Chl ligands of IsiA1 are consistent with the ligands of the structurally known IsiAs from the three cyanobacteria (Figure S5 in Nagao et al., BBA 2021). Here, we show a different amino acid ligand between Q34 of IsiA1-5 and H34 of IsiA2-1 (Supplementary Fig. 6, 7a, 8a). In addition, two Chls corresponding to a414 and a415 in IsiA1-5 are lacking in IsiA2-1, because IsiA2-1 has I144 and S309 instead of H144 and Q316 of IsiA1-5 (Supplementary Fig. 6, 7a, 8a). Thus, we can show the differences in amino-acid ligands of Chls based on the present Supplementary Fig. 6, 7a, 8a and our previous study (Nagao et al.,

BBA 2021). The variations in the amino acid ligands of Chls in the *Anabaena* IsiAs may be important for light-harvesting strategy in cyanobacteria having several *isiA* genes. To explain these contents, we added a new paragraph to the Discussion section in the revised manuscript (lines 400-407).

Our structural papers have so far shown correlation of Chls with their ligands. This is because that the Reviewers in our previous structural papers stated the importance of the Figures of Chl-ligand interactions. We agree with these Reviewer's comments and believe the importance of Figures showing the correlation of Chls with their ligands. Therefore, we would like to keep these figures in the revised manuscript.

Comment 3:

3. Fig. S7b & c, please indicate which IsiA1 is shown here, or are these two regions from two different IsiA1s?

Author reply 3:

Fig. S7b is IsiA1-4 and Fig. S7c is IsiA1-5. We added this information to the Supporting Information.

Reviewer #5 (Remarks to the Author):

Now I have read the revised manuscript of Nagao et al. and do find it greatly improved and all queries addressed.

I have no further comments and hope it will be published soon.

Merry Christmas and a Happy New Year to everyone.

Author reply:

Thank you.